# TOOL-AUGMENTED REWARD MODELING

**Lei Li**[*♡]    **Yekun Chai**[*♠]    **Shuohuan Wang**[♠]    **Yu Sun**[♠]
**Hao Tian**[♠]   **Ningyu Zhang**[♡]   **Hua Wu**[♠]
[♡]Zhejiang University    [♠]Baidu Inc.
{leili21,zhangningyu}@zju.edu.cn
{chaiyekun,wangshuohuan,sunyu02}@baidu.com

## ABSTRACT

Reward modeling (*a.k.a.*, preference modeling) is instrumental for aligning large language models with human preferences, particularly within the context of reinforcement learning from human feedback (RLHF). While conventional reward models (RMs) have exhibited remarkable scalability, they oft struggle with fundamental functionality such as arithmetic computation, code execution, and factual lookup. In this paper, we propose a tool-augmented preference modeling approach, named Themis, to address these limitations by empowering RMs with access to external environments, including calculators and search engines. This approach not only fosters synergy between tool utilization and reward grading but also enhances interpretive capacity and scoring reliability. Our study delves into the integration of external tools into RMs, enabling them to interact with diverse external sources and construct task-specific tool engagement and reasoning traces in an autoregressive manner. We validate our approach across a wide range of domains, incorporating seven distinct external tools. Our experimental results demonstrate a noteworthy overall improvement of 17.7% across eight tasks in preference ranking. Furthermore, our approach outperforms Gopher 280B by 7.3% on TruthfulQA task in zero-shot evaluation. In human evaluations, RLHF trained with Themis attains an average win rate of 32% when compared to baselines across four distinct tasks. Additionally, we provide a comprehensive collection of tool-related RM datasets, incorporating data from seven distinct tool APIs, totaling 15,000 instances. We have made the code, data, and model checkpoints publicly available to facilitate and inspire further research advancements[1].

## 1 INTRODUCTION

Large language models (LLMs) have demonstrated remarkable potential in performing complex tasks that demand expertise across diverse domains, such as programming (Chen et al., 2021; Li et al., 2022; Chai et al., 2023; Li et al., 2023) and dialogue assistance (Bai et al., 2022a; Ouyang et al., 2022; OpenAI, 2023; Anil et al., 2023; Touvron et al., 2023). Leveraging reinforcement learning from human feedback (RLHF; Christiano et al., 2017; Stiennon et al., 2020) has emerged as a compelling approach for optimizing LLMs against reward models (RMs) to predict human preferences. RMs serve as imperfect proxies for human feedback signals, producing rewards that are pivotal for fine-tuning LLMs in RLHF. However, RMs predict human preferences relying on static internal representations stored within their weights, which inherently impose limitations of LLMs. These may encompass challenges in accessing real-time information on current events (Komeili et al., 2022) and a propensity to hallucinate erroneous facts (Zhang et al., 2023), a lack of proficiency in arithmetic computation (Lewkowycz et al., 2022), and difficulties in comprehending low-resource languages (Lin et al., 2022b), among others. These limitations underscore the imperative need to engage external sources of information to augment and improve the effectiveness of RMs.

To further motivate this shift, an intriguing question arises when considering the role of human labelers in generating RM data: do these labelers possess an intuitive inclination to resort to external

---

[*]Equal contribution and shared co-first authorship.
Work done during LL's internship at Baidu. Correspondence to: SW, NZ.
[1]https://github.com/ernie-research/Tool-Augmented-Reward-Model

tools, much like humans themselves (akin to human problem-solving behavior) ? The motivation behind this question stems from the observation that even human labelers, when faced with complex tasks, often turn to external aids such as search engines or calculators to retrieve knowledge, validate facts, and facilitate their decision-making process. On the other hand, recent studies have unveiled the impressive performance gains that can be achieved by integrating external tools into the reasoning process of LLMs. Recent works such as Chain-of-Thought (Wei et al., 2022) and ReAct (Yao et al., 2023) have demonstrated that step-by-step reasoning and tool use can significantly enhance the planning and reasoning abilities of LLMs, enabling them to successfully complete intricate tasks.

In response to these insights, this paper presents `Themis`, a tool-augmented RM framework that combines tool engagement and reasoning process in a sequential and step-by-step manner. Our approach endows RMs with the capacity to make dynamic decisions regarding which APIs to call, when to invoke them, what arguments to pass, and how to effectively incorporate the results into the broader reasoning process. This approach empowers RMs to engage in dynamic reasoning, enabling it to make high-level plans for tool use (reasoning to tools) while also interacting with external environments to incorporate additional information into its reasoning (reasoning to rewards).

One crucial advantage of this framework is that it offers a significant departure from vanilla pairwise RMs, which has been inherently likened to a "blackbox" due to its opacity in revealing the internal reasoning process. In contrast, our approach provides a transparent and sequential account of actions and verbal reasoning traces specific to a given task. This transparency not only enhances human interpretability but also engenders trustworthiness, as it unveils the inner workings of the RM's decision-making process. This facilitates fine-tuning and modification of intermediate steps to exert precise control over the reward generation process.

To facilitate the exploration and validation of our proposed framework, we meticulously curated a comprehensive dataset comprising interactions with seven distinct external tools. The construction of this dataset was a collaborative endeavor, synergizing the generative capabilities of GPT-4 (OpenAI, 2023) as a prompting engine, tool-executed-based filtering, and human annotations.

We comprehensively evaluate `Themis` across a diverse spectrum of domains, encompassing the utilization of these seven distinct external tools. Experimental results demonstrate that `Themis` yields a remarkable 17.7% improvement compared to conventional RMs that lack access to external tools, across eight distinct tasks. Moreover, `Themis` outperforms Gopher 280B by a substantial margin of 7.3% on TruthfulQA benchmark, consistently surpassing baseline RMs. These compelling results underscore the effectiveness and generalization capability of `Themis` in enhancing truthfulness and factuality in preference modeling. Furthermore, we extend our investigation to RLHF fine-tuning, revealing that our method attains an impressive 32% win rate on average across four different tasks when compared to vanilla RMs, as determined through human preference evaluation. This further demonstrates the practical utility and superiority of our approach in real-world applications.

To summarize, our key contribution are encapsulated as follows: (1) We advance the domain of tool-augmented preference modeling by introducing the `Themis` framework. This framework harnesses the power of *external tools* to augment preference modeling in LLMs. In doing so, it mitigates inherent limitations observed in conventional RMs. Additionally, our approach unveils the inner workings of the RM's decision-making process, providing transparency and interpretability. (2) We present a novel tool-augmented reward modeling dataset (TARA) that includes comprehensive comparison data of human preferences and detailed tool invocation processes. This dataset will be made publicly available in hopes of facilitating research advancements in the field. (3) Our contributions are substantiated through experimental evaluations conducted across eight diverse tasks within TARA, as well as benchmarking against TruthfulQA and Retarded-bar datasets. These experiments conclusively demonstrate the effectiveness of our approach in enhancing the performance of LLMs.

## 2 TOOL-AUGMENTED REWARD MODELING

### 2.1 REVISITING REWARD MODELS

In RLHF (Ouyang et al., 2022; Stiennon et al., 2020), RM is trained on a human preference dataset consisting of comparisons between two candidate model outputs generated in response to the same input prompt. The vanilla RM operates by mapping each input prompt and its corresponding gen-

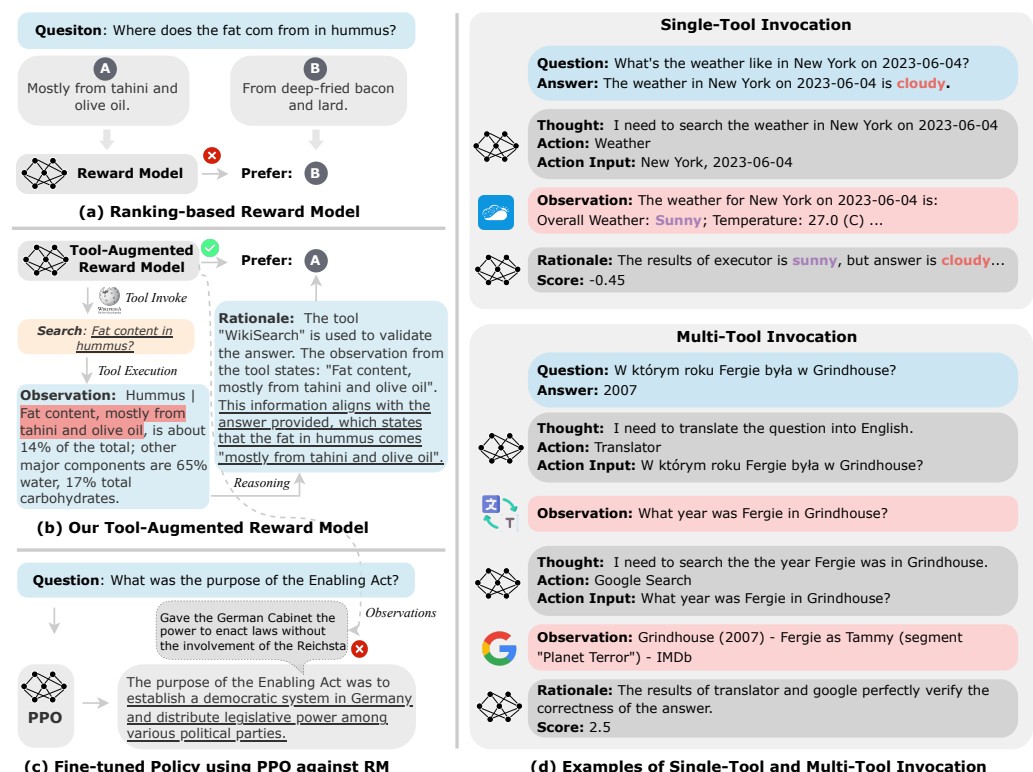

Figure 1: A diagram illustrating the pipeline of (a) Vanilla reward models (RMs); (b) Tool-augmented RMs, namely `Themis`; (c) Reinforcement learning via proximal policy optimization (PPO) on above RMs; (d) Examples of single or multiple tool use process in the proposed approach. See Section 2 for more details of our method.

erated output to a scalar reward, thereby encapsulating the overall preference between them. Mathematically, for a given question denoted as $x$ with a positively preferred answer represented as $y_w$ and a negatively preferred answer as $y_l$, the loss function of the vanilla RM is formulated as:

$$\mathcal{L}_{\text{RM}} = -\mathbb{E}_{(x,y_w,y_l)\sim D}[\log(\sigma(r_\theta(x, y_w) - r_\theta(x, y_l)))] \tag{1}$$

Here, $r_\theta(x, y)$ represents the scalar output of the RM for question $x$ and answer $y$, $\sigma$ denotes sigmoid function, while $D$ denotes the preference dataset. However, as highlighted earlier, this vanilla RM approach, while useful in aligning reward scores with human preferences, is constrained by limitations pertaining to timeliness and knowledge accessibility. It faces particular challenges when confronted with complex tasks such as mathematical problem-solving and reasoning.

## 2.2 THEMIS: TOOL-AUGMENTED REWARD MODELING

Figure 1 presents an overview of the `Themis` framework, illustrating how it integrates tool engagement and reasoning processes in a structured, step-by-step manner. Our approach enhances RMs with the capability to make informed and dynamic decisions concerning which APIs to employ, when to invoke them, what arguments to pass, and how to effectively integrate the obtained results into the broader reasoning process. This comprehensive framework, depicted through step-by-step trajectories, encapsulates the entirety of the decision-making and reasoning journey, consisting of the following pivotal stages:

- **Thought**: At this initial stage, the model evaluates whether it should engage external APIs (referred to as tool reasoning).
- **Action**: Subsequently, the model generates the necessary API calls along with the corresponding arguments required for the interactions.
- **Observation**: The results produced by the external APIs are collected and stored.
- **Rationale**: This stage involves the aggregation and synthesis of previously acquired information, fostering both induction and reasoning processes, specifically tailored for reward modeling.

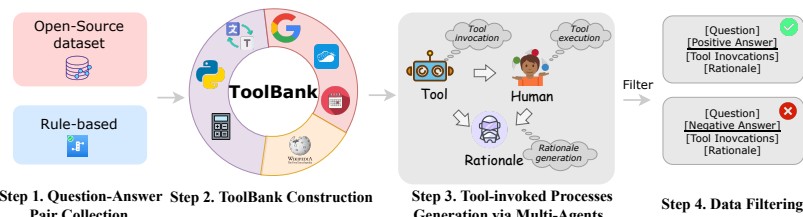

Figure 2: An illustration of data creation pipline for our **T**ool-**A**ugmented **D**at**A**set (TARA).

- **Reward**: Finally, the model leverages the accumulated insights and information to generate a scalar reward score with a feed-forward layer, reflecting its overall preference based on the collective evidence.

Given a question $x$ and the corresponding generated output $y$, we consider a generalized RM agent parameterized by $\theta$ with the capability to interact with external tools. Following Yao et al. (2023), we define the agent's action state $a_t \in \mathcal{A}$ at step $t$ as a combination of a natural language thought $\hat{a}_t$ and a tool acting state $\bar{a}_t \in \mathcal{T}$. These paired "(thought, action)" states are denoted as $a_t = (\hat{a}_t, \bar{a}_t)$. The purpose of the thought component $\hat{a}_t$ is to encapsulate the comprehension of pertinent information and to guide the ensuing action $\bar{a}_t$. This action is determined following a policy $p_\theta(\bar{a}_t|x, y, c_t, \hat{a}_t)$, where $c_t = (a_1, o_1, \cdots, a_{t-1}, o_{t-1})$. Subsequently, the RM agent is tasked with predicting a reasoning thought based on the preceding context denoted as $s_T$. This reasoned thought *Rationale* plays a pivotal role in enhancing the model's ability to summarize and reason effectively, drawing upon the historical reasoning traces before ultimately predicting the scalar reward $r$. This approach closely mirrors the step-by-step reasoning paradigm established by Chain-of-Thought (Wei et al., 2022), accentuating the incremental and interactive nature inherent in the RM's decision-making process. Formally, the complete reasoning trajectory is represented as $c_{1...T} = (a_1, o_1, \cdots, a_T, o_T, s_T)$. Consequently, the reward can be denoted as $r_\theta(x, y, c_{1...T})$. During the training phase, we implement an auto-regressive training objective for the prediction of the next token in modeling the reasoning context $c_t$. In the context of reward training, we produce a scalar reward based on $c_t$ by a fully connected layer and employ the same pair-wise ranking loss as utilized in conventional RMs. This loss function serves as a foundational component to discern and rank the relative preferences between different model-generated outputs. All stages, with the exception of the **Reward** stage, utilize language model heads to generate text tokens (same as language models). In the **Reward** stages, a real-valued score is produced using a feed-forward layer (same as conventional RMs).

**Training Objectives**  The overall training objective is comprised of two distinct components: the *pair-wise ranking loss* and the *auto-regressive language modeling loss*. The former aligns with equation 1, while the latter is designed to empower RMs with the ability to perform tool invocation via supervised fine-tuning:

$$\mathcal{L}_{\text{total}} = \underbrace{\mathcal{L}_{\text{RM}}}_{\text{pair-wise ranking loss}} + \underbrace{\alpha\big(\sum_{t=1}^{T}(\mathcal{L}_{\text{tool}(t)} + \beta\mathcal{L}_{\text{Observation}(t)}) + \omega\mathcal{L}_{\text{Rationale}}\big)}_{\text{auto-regressive language modeling loss}} \qquad (2)$$

where $\beta, \omega \in \{0, 1\}$ are hyper-parameters to modulate different training configurations, $T$ represents the number of tool invocations, $\alpha = 1$ in our experiments. Please refer to Table 8 for details.

**Connection to Vanilla RM**  When $\alpha$ is set to zero, the autoregressive loss term becomes null, effectively reducing `Themis` to standard RMs. Ideal RMs should possess the ability to discern when and whether to employ external tools. To impart this knowledge, we incorporate tool-related data alongside non-tool data, thereby instructing RMs on the appropriate timing for tool invocation. Notably, this framework inherently encompasses the functionality of vanilla RMs.

## 3 TOOL-AUGMENTED REWARD DATASET

### 3.1 DATA COLLECTION

The comprehensive construction process of the **T**ool-**A**ugmented **R**eward d**A**taset (**TARA**) is depicted in Figure 2. TARA is constructed by leveraging high-quality datasets and generating the

tool invocation process through multi-agent interactions. This process can be subdivided into the following four key steps:

**Step 1: Question-Answer Pairs Collection** Initially, we collect a reward dataset featuring each instance comprising a question, a positive answer, and a negative answer. To construct this dataset, we employ two distinct approaches: resume open-source, high-quality datasets, and generation from scratch with some heuristic methods. However, the above methods usually only produce positive answers. To address this concern, we leverage GPT-4 as a negative generation agent to generate antagonistic negative answers, which will be described in Step 3.

**Step 2: ToolBank Construction.** Subsequently, we develop the toolbank, which encompasses three distinct types of tools: basic tools, query-based tools, and knowledgeable tools. Basic tools such as *Calculator*, *Code Interpreter*, and *Translator*, provide RMs with practical capabilities. Query-based tools, including *Google*, *Weather*, and *Calendar Search*, equip RMs with search capabilities to access up-to-date information. Knowledgeable tools enable RMs to tap into a knowledge base, enhancing the factual accuracy of rewards. Additionally, we introduce Multi-Tools, which contain sequential calls to multiple tools.

**Step 3: Tool-invoked Process Generation by Multi-Agents.** To automate the generation of tool-invoked processes, we design a simulated environment featuring human participants and three virtual agents: the negative generation agent, the tool agent, and the rationale agent. Leveraging the substantial comprehension and generation capabilities of LLMs, we employ three GPT-4 to simulate the roles of the three agents. The negative generation agent is responsible for generating a negative answer to the question that has only a positive answer. It processes a question along with its positive answer and generates a negative answer that is indistinguishable from the positive one. The tool agent acts as an intermediary between humans and agents, deciding when, which, and how to invoke tools. Specifically, the tool agent receives a question-answer pair and produces **Thought** and **Action** stages, as outlined in Section 2.2. Then humans are tasked with invoking specific tools and observing the outcomes (**Observation** stage). The rationale agent is tasked with generating reasoning traces by comprehending previous contexts, synthesizing the question-answer pair, the tool invocation process, and the observations to systematically produce rewards (**Rationale** stage). Tool-invoked scenarios are simulated through interactions between agents and humans, yielding tool-invoked processes for RMs. The interaction process and prompts for each agent are detailed in Appendix Figure 13.

**Step 4: Data Filtering.** To maintain data quality, we implement a straightforward yet effective filtering process on the generated data. Initially, we exclude tool-invoked processes acquired in Step 3 that exhibit invalid formats. Subsequently, we discard processes exceeding three interaction steps, lacking relevant function calls, or manifesting parsing errors in their execution results.

## 3.2 DATA STATISTICS

The data statistics and comparison to previous datasets are shown in Appendix Table 5. TARA comprises a total of 13,604 training datasets and 1,469 test sets, each consisting of a question, a positive answer, and a negative answer. TARA incorporates a diverse set of seven tools that span across various domains, encompassing mathematical operations, code-related inquiries, closed-ended and open-ended question answering, knowledge-based queries, and time-sensitive information requests.

## 4 EXPERIMENTS

### 4.1 EXPERIMENTAL SETTINGS

We utilized Vicuna-7B (Zheng et al., 2023) as the base model for `Themis` and compared it with conventional RMs, specifically Bert-Large (Devlin et al., 2019) and Vicuna-7B, denoted as RM (Bert-Large) and RM (Vicuna-7B) respectively, which serve as its underlying architectures. Our training and evaluation processes were conducted under two distinct settings: single-tool and mixed-tool scenarios. In the single-tool configuration, the TARA data was partitioned based on tool types, with each model exclusively trained on a specific type of tool. In contrast, the mixed-tool scenario involved training models on the entirety of the TARA dataset, encompassing diverse tools. During evaluation, we compared the relative reward scores for positive and negative answers, using accu-

Table 1: The main results on the Tool-Augmented Reward Dataset (TARA). We report the performance of RM and Themis in both single-tool and mixed-tool settings. **Bold** scores highlight the best performance achieved. The reported **Avg.** values are calculated by averaging accuracy across all instances, offering a comprehensive measure of micro accuracy that spans various tool types.

| Model | Calendar | Calculator | Weather | Code | Translator | Wiki | Google | Multi | Avg.↑ |
|---|---|---|---|---|---|---|---|---|---|
| *single-tool setting* | | | | | | | | | |
| RM (Bert-Large) | 63.21 | 88.31 | 71.52 | 66.67 | 24.33 | 82.75 | 68.66 | 78.47 | 65.01 |
| RM (Vicuna-7B) | 80.91 | 98.05 | 86.08 | 85.19 | 34.33 | 93.31 | 65.13 | 79.17 | 75.04 |
| Themis (Vicuna-7B) | **100.00** | **98.70** | **100.00** | **99.47** | 88.40 | 95.07 | **76.12** | **99.31** | 94.23 |
|   w/o $L_{\text{Observation}}$ | **100.00** | 98.05 | **100.00** | **99.47** | 87.71 | 90.49 | 64.48 | 80.56 | 90.23 |
| *mixed-tool setting* | | | | | | | | | |
| RM (Bert-Large) | 83.02 | 94.16 | 80.38 | 73.54 | 22.67 | 83.45 | 70.15 | 81.25 | 69.10 |
| RM (Vicuna-7B) | 83.96 | 94.16 | 83.54 | 88.36 | 33.67 | 92.61 | 72.39 | 81.25 | 75.63 |
| Themis (Vicuna-7B) | **100.00** | 98.05 | **100.00** | **99.47** | 90.91 | 93.31 | 64.92 | **99.31** | 93.31 |
|   w/o $L_{\text{Observation}}$ ($\beta = 0$) | **100.00** | 98.05 | **100.00** | **99.47** | 91.47 | 94.37 | 62.69 | 73.51 | 90.90 |
|   w/o $L_{\text{Rationale}}$ ($\omega = 0$) | **100.00** | 96.75 | 99.37 | 98.94 | 88.74 | 92.54 | 63.43 | 68.72 | 89.31 |
| Themis (Vicuna-33B) | **100.00** | 97.40 | **100.00** | **99.47** | **93.54** | **96.55** | 73.72 | **99.31** | **95.21** |
| Themis (Vicuna-7B + LoRA) | 96.22 | 96.10 | 96.20 | **99.47** | 73.33 | 90.49 | 46.26 | 58.33 | 82.57 |
| Themis (Vicuna-13B + LoRA) | 98.11 | 92.21 | 98.73 | 98.41 | 72.00 | 92.25 | 57.85 | 75.69 | 85.26 |
| Themis (Vicuna-33B + LoRA) | 86.79 | 97.40 | 99.36 | 98.41 | 84.66 | 95.77 | 58.95 | 99.30 | 90.74 |

Figure 3: **Left**: Model performance for various training epoch numbers; **Right**: Visualization of the change of average reward scores with training epochs. The top reward score line of each model corresponds to the positive answer, while the bottom line corresponds to the negative answer.

racy as the evaluation metric. Further details about hyper-parameter choices and additional training specifics can be found in Appendix C.1.

## 4.2 MAIN RESULTS

**Single-Tool vs. Mixed-Tool Performance.** The main performance results of all models on TARA are shown in Table 1. Across both the single-tool and mixed-tool settings, it is evident that Themis consistently outperforms vanilla RMs significantly, exhibiting an improvement of +19.2% in the single-tool scenario and +17.7% in the mixed-tool context across 8 distinct tasks. Enabling access to external knowledge and information, specific tools significantly boost the performance of *Calendar* (+19.1%), *Weather* (+13.9%), *Code* (+14.3%), and *Translator* (+54.1%) respectively. Remarkably, Themis achieves a perfect accuracy of 100% on *Calendar* and *Weather* tasks. Moreover, it attains an accuracy above 98% on *Code* and *Calendar* tasks, providing substantial evidence for the efficacy and motivation behind integrating external tools into the reasoning process.

In mixed-tool experiments, Themis demonstrates robust performance in concurrently learning diverse tools, with 7 out of 8 tasks displaying superior performance compared to the baselines. Notably, *Google Search* exhibits a slight decline during mixed-tool training. This can be attributed to the diverse sources from which data is collected, especially from Reddit, resulting in overlapping domains and complexity for the models to learn effectively. Addressing this challenge necessitates meticulous data cleaning and rigorous human annotation processes for further exploration. These findings underscore the remarkable potential of Themis across a broader spectrum of tools, emphasizing its adaptability and versatility in real-world applications.

**Scaling Trends in Themis.** In our investigation, we explored models spanning different scales, ranging from Vicuna-7B to Vicuna-33B, all within the context of a mixed-tool setting. To enhance the efficiency of our training, we utilized the LoRA technique (Hu et al., 2022) for parameter-

| Model | #Param | Zero-shot | Fine-tuning |
|---|---|---|---|
| RM (Bert-Large) | 340M | 51.66 | 52.50 |
| RM (Vicuna-7B) | 7B | 35.78 | 65.83 |
| Themis | 7B | 55.00 | 70.00 |
| w/o $L_{\text{observation}}$ | 7B | **55.83** | **71.67** |

Table 2: Results on the HH-RLHF* dataset, comparing Themis with vanilla RMs in zero-shot and finetuning evaluation.

| Model | #Param | TruthfulQA↑ | Retarded-bar (en)↑ |
|---|---|---|---|
| GPT-3 | 175B | 21.0 | - |
| OPT | 175B | 21.0 | - |
| Gopher | 280B | 29.5 | - |
| Galactica | 120B | 26.0 | - |
| RM (Vicuna) | 7B | 30.7 | 68.0 |
| Themis | 7B | **36.8** | **73.3** |

Table 3: Results on TruthfulQA (MC1) and Retarded-bar datasets.

efficient fine-tuning. The experimental results, outlined in Table 1 (last 3 rows), elucidate a positive correlation between the scale of the model and its overall performance, a phenomenon that aligns with established scaling laws (Gao et al., 2023; Askell et al., 2021). We also full-parameter fine-tune Themis (Vicuna-33B) and obtain the best performance.

**Effect of Varying Training Epochs.** As depicted in Figure 3 (left), we observe that RM (Bert-Large) necessitates more training epochs to reach convergence, whereas RM (Vicuna-7B) achieves optimal performance after just a single epoch—an observation consistent with prior research findings (Stiennon et al., 2020; Wu et al., 2021; Ouyang et al., 2022). In contrast, Themis does require additional training epochs to learn tool invocations and rewards effectively. However, it outperforms traditional RMs, even within a single training epoch.

**Reward Difference Visualization.** We visualize the average reward scores vary with the number of training epochs in the right of Figures 3. Themis consistently exhibits a proclivity to assign higher scores to positive answers and lower scores to negative answers, indicating a heightened level of differentiation. Moreover, this differentiation progressively intensifies as the model training.

## 4.3 ANALYZING THE ROLE OF TOOL USE

**How Does Themis Learn to Use Tools?** To understand the role of the tools and how the RMs learn the tool invoking, we analyze the relationship between the number of tool invocations, model performance, and training epochs as shown in Figure 4. Our findings indicate a pattern in the total number of tool calls, which initially increases and then decreases. Additionally, we observe a gradual reduction in the number of incorrect tool calls. These trends collectively suggest that **Themis acquires the ability to invoke tools effectively**. As depicted on the right side of Figure 4, our observations reveal that Themis consistently exhibits a higher propensity to utilize *Google Search* during problem-solving tasks. This aligns well with human behavior, reflecting the natural inclination of individuals to resort to search engines when seeking solutions.

**Does Themis Really Make Decisions Based on Observations?** We specifically selected data instances where Themis effectively differentiated between positive and negative answers. Subsequently, we manipulated the outcomes of its tool invocations (*Observations*) and adjusted its states to assess Themis's performance. For instance, let's consider the question "What is the weather like in New York on 2023-06-24?". Both the positive answer and the initial Weather observation were recorded as "Sunny". We modified the observation to "Raining", thus transforming the answer into a negative response. Remarkably, we observed only a marginal decrease in accuracy, highlighting the robust alignment between our method's predictions and the tool observations.

**Ablation Study: Do Tool Use and Reasoning Traces Count?** To comprehend the functionalities of the reasoning process of Themis, we set $\beta = 0$ and $\omega = 0$ to mask *Observation* and *Rationale* in Themis, respectively. The results can be seen in Table 1, highlighting the substantial contributions of both *Observation* and *Rationale* to Themis, especially in the *Multi-Tools* category. We find that the performance of Themis clearly drops when we exclude the *Rationale* component, proving the effectiveness of step-by-step reasoning before output reward scores.

**Case Study.** We show some qualitative examples in Appendix D.1, showcasing the effectiveness of the tool invocations. By leveraging the *external tools*, Themis validates the answer accuracy and make decisions through a systematic, step-by-step reasoning process.

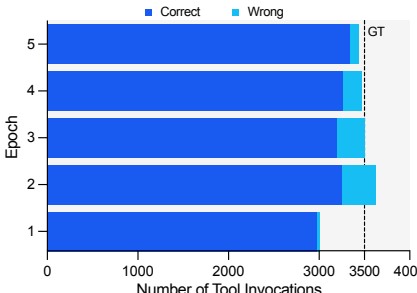 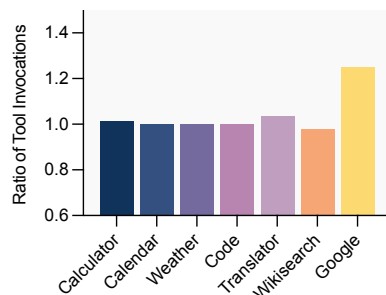

Figure 4: **Left**: The variations in the number of correctly invoked tools and incorrectly invoked tools. The dashed line is the total number of invoked tools in TARA. And the pentagram refers to the best performance epoch. **Right**: Comparison of the number of invoked different tools.

## 4.4 GENERALIZATION PROBING IN DOWNSTREAM TASKS

**Out-of-Domain Evaluation.** Ideally, `Themis` is expected to possess adaptive tool invocation capabilities and the ability to score unseen prompts and responses. Consequently, we select 150 instances (one instance has one positive answer and one negative answer) from HH-RLHF (Bai et al., 2022b) and denote it as HH-RLHF*. Initially, we assess the performance of both RMs and `Themis` (after training on our TARA dataset) in the zero-shot setting to estimate their extrapolation ability. As shown in Table 2, our findings reveal that `Themis` consistently outperforms all RMs, especially RM-Vicuna-7B. We further introduce 500 instances for model finetuning, and we find the effect of `Themis` is significantly improved. It indicates that our `Themis` can extrapolate to output-of-domain scenarios effectively with minimal data fine-tuning.

**More than RM: Truthfulness and Factuality Probing.** Given that RMs are employed to rank various responses for a single prompt, it is natural to leverage RMs for addressing multiple-choice tasks. We experimented on multiple-choice problems on TruthfulQA (Lin et al., 2022a) and translated Retarded-bar[2], denoted as Retarded-bar (en), to access the model's capacity for truthfulness and factuality (Refer to Appendix D.2 for dataset details). As shown in Table 3, `Themis` outperforms competitive LLMs including OPT 175B (+15.8%), GPT-3 (+15.8%), Galactica (+10.8%), and Gopher 280B (+7.3%) on TruthfulQA task in zero-shot evaluation. Moreover, we selected Retarded-bar, a challenging dataset containing puns, unusual punctuation, and irrational logic, to assess `Themis`'s ability in solving fact-related confusing problems. We show that `Themis` overshadows the vanilla RM on Retarded-bar (en) by 5.3%. Examples in Appendix D.2 showcase that `Themis` can retrieve knowledge with external tools and enhance its truthfulness capability.

## 4.5 EXTENDED EXPERIMENTS: FROM RLHF TO RLTAF

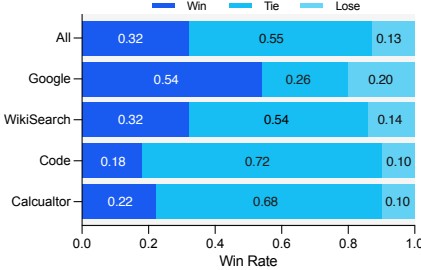

Figure 5: Human preference evaluation, comparing PPO (`Themis`) to PPO (vanilla RM) across 200 test prompts.

| Model | PPL ↓ |
|---|---|
| Vicuna-7B | 11.19 |
| Vicuna-7B-SFT | 8.14 |
| Vicuna-7B-PPO (RM) | 8.10 |
| Vicuna-7B-PPO (`Themis`) | **7.88** |

Table 4: The perplexity evaluation in RLHF across different stages in PPO, SFT, *etc*. Our model outperforms base model, SFT model, and PPO with conventional RMs.

**Automatic Evaluation.** We conducted experiments to assess the impact of `Themis` in RLHF, namely reinforcement learning from tool-augmented feedback (RLTAF). Following prior studies (Stiennon et al., 2020; Ouyang et al., 2022), we implemented three stages: supervised fine-tuning (SFT), traditional RMs, and fine-tuning the policy against these RMs using Proximal Policy Optimization (PPO). Utilizing TARA as our training data, detailed experimental specifics can be found

---

[2]`https://huggingface.co/datasets/hugfaceguy0001/retarded_bar`

in Appendix C.2. In RLHF, `Themis` employs external tools for tasks like arithmetic computation, code execution, and factual lookup. The results presented in Table 4 indicate that PPO optimized against `Themis` achieves lower perplexity compared to vanilla RMs.

**Human Preference Evaluation.** We further conducted human evaluation, comparing win:tie:lose ratios across four domains on 200 test prompts. As illustrated in Figure 5, our method outperforms baselines, achieving an average +32% win rate across the four different domains and consistently surpassing vanilla RMs in all four tasks. Notably, our approach demonstrated substantial improvements in fact-related question answering and arithmetic computation, providing robust evidence for the effectiveness of our methodology.

## 5 RELATED WORK

### 5.1 REWARD MODELING IN HUMAN ALIGNMENT

The challenge of aligning machine learning systems with human preferences has seen considerable exploration. Early efforts involved training aligned agents by imitating human behavior (Pomerleau, 1991; Abbeel & Ng, 2004; Ho & Ermon, 2016; Finn et al., 2016). However, these methods often struggled to outperform humans due to the requirement for substantial amounts of expensive data. A scalable solution was proposed by Leike et al. (2018), utilizing an RM to align machine learning systems with human performance. Stiennon et al. (2020) fine-tuned language models through reinforcement learning by training a RM to mimic human preferences in summarization tasks. Similar approaches were adopted by Nakano et al. (2021); Ouyang et al. (2022); Bai et al. (2022b), focusing on aligning LLMs like GPT-3 towards honesty, helpfulness, and harmlessness. In a parallel vein, Shen et al. (2023a); Le et al. (2022); Bukharin et al. (2023); Shojaee et al. (2023) contribute to the field by focusing on reward design in reinforcement learning for code-related tasks, which is pivotal for augmenting the code understanding and generation capabilities of models. However, these existing RMs faced significant challenges, such as real-time information processing, limited to specific tasks such as summarization and code generation, and struggle with assigning rewards for intricate tasks like mathematics., To address these limitations, our approach incorporates external tools, augmenting the reward model and mitigating these challenges.

### 5.2 TOOL LEARNING

The intersection of specialized tools and LLMs has recently gained significant attention in research (Mialon et al., 2023; Qin et al., 2023). Current studies in this area can be categorized into two main approaches: tool-oriented learning (Yao et al., 2023; Nakano et al., 2021; Qian et al., 2023; Shen et al., 2023b) and tool-augmented learning (Schick et al., 2023; Lu et al., 2023; Tang et al., 2023; Qiao et al., 2023). In tool-oriented learning, LLMs serve as decision-making hubs for the strategic use of tools. Conversely, tool-augmented learning treats tools as complementary resources that enhance LLMs' capabilities. Unlike previous works, our focus lies in tool-augmented reward modeling, aiming to align rewards with human preferences by incorporating tool assistance.

## 6 CONCLUSIONS AND FUTURE WORK

In this paper, we introduce `Themis`, a novel approach designed to enhance reward models by enabling interaction with *external tools*, thereby facilitating a step-by-step reasoning trajectory. Our contribution also includes the creation of a comprehensive tool-augmented reward dataset, TARA, which encompasses detailed data on human preferences and intricate tool invocation processes. Through comprehensive experimentation, including preference ranking analysis, ablation studies, generalization assessments, and RLHF/RLTAF probing, we have demonstrated the substantial benefits of `Themis` in augmenting interpretive capacity and scoring reliability. Our results underscore the effectiveness of our approach in integrating external tools seamlessly into the reward modeling process. Looking ahead, an exciting avenue for future research could involve exploring `Themis` in multi-turn dialogue generation. Understanding the intricate dynamics between external tools and natural language generation in interactive conversational contexts presents a promising and intriguing direction for further investigation.

## ACKNOWLEDGMENTS

We would like to express our gratitude to the anonymous reviewers for their insightful and constructive feedback. Additionally, our appreciation goes to Yaqian Han, Pan Tang, and Yuchen Ding for their helpful assistance with the human evaluation process. This work was supported by the National Natural Science Foundation of China (No. 62206246), the Fundamental Research Funds for the Central Universities (226-2023-00138), Zhejiang Provincial Natural Science Foundation of China (No. LGG22F030011), Ningbo Natural Science Foundation (2021J190), Yongjiang Talent Introduction Programme (2021A-156-G), CCF-Baidu Open Fund, and Information Technology Center and State Key Lab of CAD&CG, Zhejiang University.

## REPRODUCIBILITY STATEMENT

To ensure the reproducibility of our experiments detailed in Section 4, we are committed to fostering transparency and accessibility in our research methodology. The source TARA datasets, fundamental to our study, have made publicly available on GitHub[3]. In addition to datasets, we provide comprehensive support for replicating our experiments. The detailed source code for both the conventional Reward Model (RM) and our proposed `Themis` approach is included in the supplementary materials. These materials encompass all necessary scripts and hyper-parameters essential for reproducing our results. The availability of datasets and thorough documentation of our experimental setup underline our dedication to promoting rigorous and replicable scientific research.

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

## A    LIMITATIONS

**Limited Tool Scope.**    While our study incorporated experiments with seven distinct tools, the po-tential of tool-augmented Reward Models (RMs) could further be explored by expanding the range of tool APIs to encompass hundreds or even thousands of diverse tools. Additionally, integrating with interfaces like the ChatGPT plugin could offer a broader application spectrum.

**Time Overheads.**    The integration of external tools introduces an additional cost of complexity and might result in increased processing time. The speed of tool invocation is contingent on various factors such as network conditions and tool execution speed. Consequently, the real-time applica-bility of the tool-augmented RM is influenced by these contextual variables.

**Single-Turn Data.** Our preference data collection was limited to single-turn prompt-response pairs. Extending this framework to encompass multi-turn interactions could enrich the understanding of complex dialogues and enhance the applicability of tool-augmented RMs in real-world conversational contexts.

**Challenges in Data Generation.** While we construct an automatic pipeline for dataset construction, we encounter certain challenges. We design some heuristic rules to generate data for particular tools, incurring associated costs that rise when extended to a broader range of tools. Additionally, we employ GPT-4 as agents to generate tool invocation processes and rationales, incurring a monetary expense associated with this computational process.

**Limited Model Scale.** Our experiments primarily revolved around Vicuna-7B. Exploring the scalability of tool-augmented RMs to models surpassing 100 billion parameters could provide insights into the challenges and opportunities at larger scales, expanding the applicability of this approach.

**Preliminary RLHF Exploration.** The exploration of tool-augmented RMs in Reinforcement Learning from Human Feedback (RLHF), namely Reinforcement Learning from Tool-Augmented Feedback (RLTAF), remains at its preliminary stages. Comprehensive experiments, covering a wider array of tasks and scenarios, are essential to fully understand the potential and limitations of this approach in reinforcement learning paradigms. Future research endeavors will focus on conducting in-depth and extensive evaluations to delve deeper into the capabilities of tool-augmented RMs in various RLHF settings.

# B  ADDITIONAL DATASET INFORMATION

Table 5: Comparison between our TARA and previous reward datasets. Our dataset contains multiple domains with tool invocations, and we construct the data via multi-agent interaction.

| Name | # Train | # Test | Domain | # Tools | Source |
|------|---------|--------|--------|---------|--------|
| WebGPT Comparisons (Nakano et al., 2021) | 19.6k | - | Long-form QA | ✗ | ELI5 & Human |
| RM-Static (Dahoas, 2023) | 76.3k | 5.1k | Helpful & Harmless | ✗ | HH-RLHF |
| Summarize from Feedback (Stiennon et al., 2020) | 179k | 6.31k | Summary | ✗ | Human |
| TARA (Ours) | 13.6k | 1.4k | Multiple | 7 | Multi-Agent |

In this section, we describe in detail the construction process of our tool-augmented reward dataset. We first introduce the generating process of each tool, then we report the details of the multi-agent prompts. Table 5 shows the data statistics of our TARA and the overview information such as source data, the number of the train-set and test-set of each tool is shown in Table 6.

Table 6: Data statistics of ToolBank.

| Tool Name | Data Source | # Train | # Test |
|-----------|-------------|---------|--------|
| Calculator | GSM-8k | 877 | 154 |
| Code | HumanEval + mbpp | 486 | 189 |
| Translator | MLQA | 1682 | 300 |
| Google Search | WebGPT Comparison | 3932 | 134 |
| Calendar | - | 320 | 166 |
| Weather | - | 476 | 158 |
| WikiSearch | NaturalQuestion | 5399 | 284 |
| Multi Tools | Calendar + Weather | 432 | 144 |

## B.1  DETAILS OF DATASET CONSTRUCTION

In this section, we introduce the details of the dataset construction. We construct the dataset based on the heuristic rule-based method and some open-source datasets with multi-agent interaction. We list instances of each tool from Figure 6 to Figure 12.

### B.1.1 Construction with Heuristic Rule-based Method

We design some heuristic rules to generate the question-answer and tool invocation processes for certain tools, including *Calendar*, *Weather*, and *Multi Tools*.

Table 7: The city set, date set, weather set, question and answer prompts set to construct the Weather tool data.

| Key | Candidate Set |
|---|---|
| City | Mexico, Saint Helier, Bangalore, Beijing, New York, Sydney, Aleppo, Homs, Sanaa, ... |
| Date | 2023-06-19, 2023-06-20, 2023-06-21, 2023-06-22, 2023-06-25, 2023-06-28, ... |
| Weather | overall weather, temperature, precipitation, humidity, wind speed, visibility, UV index |
| Question Prompts | What is the {weather} in {city} on {date}? |
| Answer Prompts | The {weather} in {city} on {date} is {answer}. |
| | {city}'s {weather} on {date} is {answer}. |
| | On {date}, {city}'s {weather} indicates {answer}. |
| | ... |

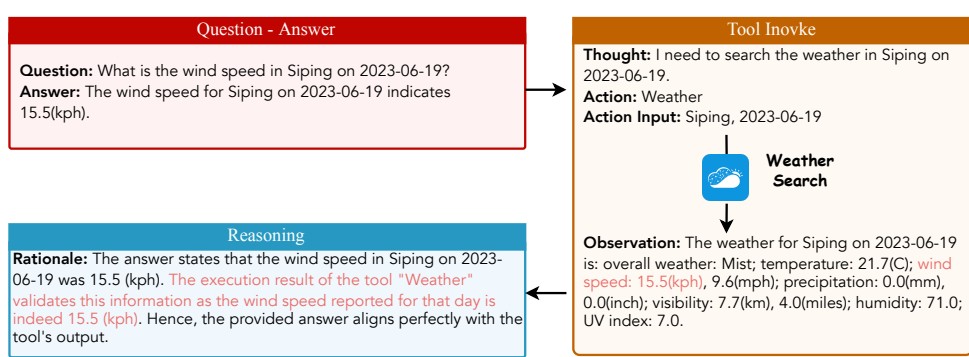

Figure 6: An example of the Weather tool.

**Weather.** The *Weather* tool is realized by `weatherAPI`[4] that the input consists of a city and a date, and the output provides information about the weather in the specified city on the given date. Initially, we compile a candidate city set by selecting the most common cities from Wikipedia. Subsequently, we create a candidate date set and a weather set specific to the Weather tool. Inspired by Wang et al. (2023), we initiate the process with a seed question prompt and an answer prompt, such as "*Question:* What is the {weather} in {city} on {date}? *Answer:* The {weather} in {city} on {date} is {answer}." and expand upon it using ChatGPT. Finally, we iterate through the city set, date set, and weather set, incorporating them into the prompts to construct the question-answer pair. We seek positive answers from the *Weather* tool, while negative answers are perturbed based on the positive answer. The city set, date set, weather set, question and answer prompts set are listed in Table 7.

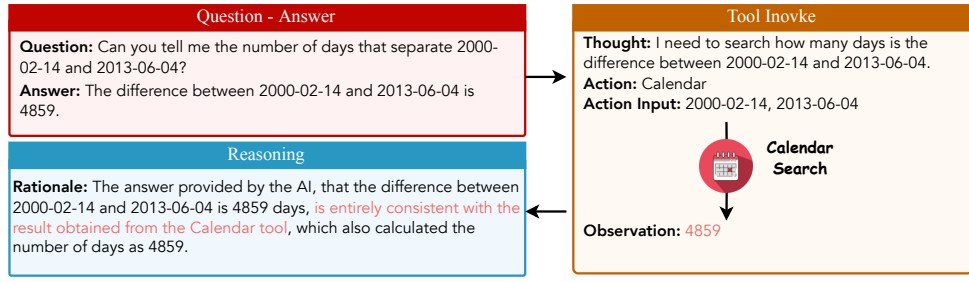

Figure 7: An example of the Calendar tool.

---

[4] `www.weatherapi.com`

**Calendar.** The construction process of the *Calendar* tool is similar to the *Weather* tool, with the primary difference lying in the question prompts and answer prompts. The *Calendar* tool serves three primary functions: determining the weekday of a given date, calculating the difference between two dates, and finding the date that follows another date by $n$ days. For each of these functions, we have composed distinct seed prompts and subsequently expanded upon them using ChatGPT.

**Multi Tools.** The *Multi-Tools* primarily involve chained invocations of the *Calendar* and *Weather* tools. An illustrative question might be, "What is the weather like in city in the $n$ days after date?" This necessitates first invoking the *Calendar* tool to obtain "the $n$ days after date" and subsequently invoking the Weather tool to retrieve "the weather like in city" for the specified date. Obviously, the data generation process for *Multi-Tools* follows the same pattern as the *Weather* and *Calendar* tools.

### B.1.2 CONSTRUCTION FROM OPEN-SOURCE DATASETS

Some tools are challenging to create using heuristic rule-based methods. Therefore, we generate them based on some open-source and high-quality datasets.

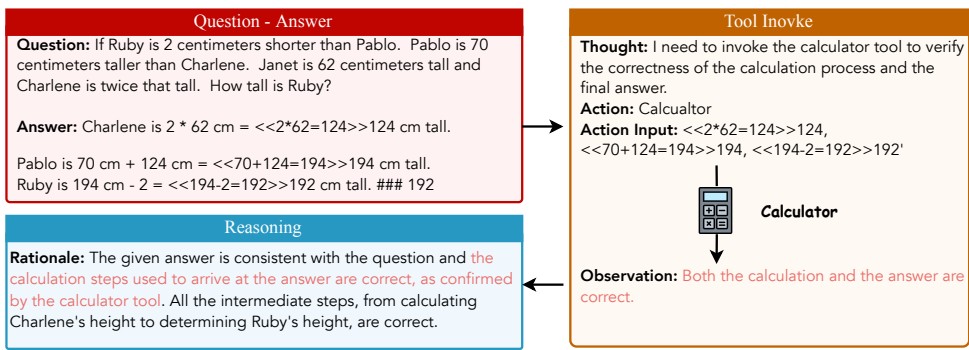

Figure 8: An example of the Calculator tool.

**Calculator.** GSM-8K (Cobbe et al., 2021) is a high-quality dataset comprising linguistically diverse grade school math word problems, making it well-suited for constructing math reasoning data involving tool invocation. We randomly selected 1,000 instances from GSM-8K to serve as both questions and positive answers. We query the negative generation agent to generate negative answers.

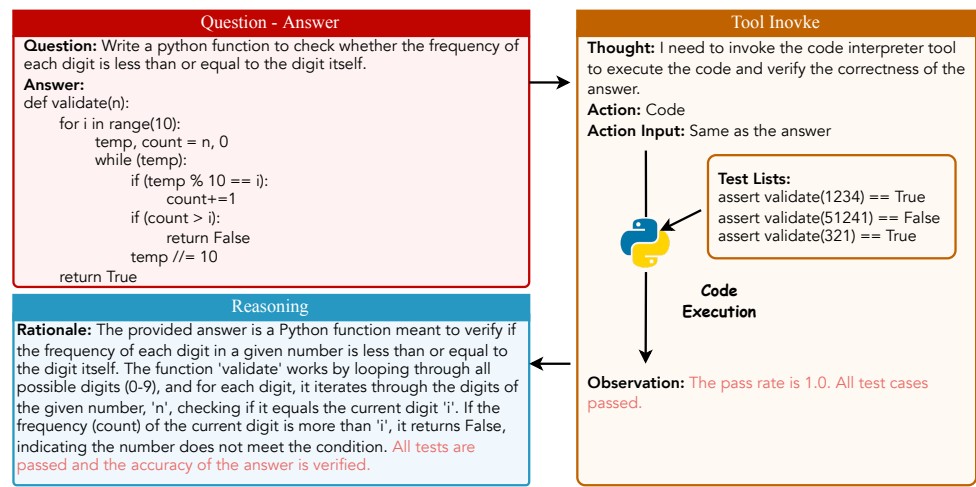

Figure 9: An example of the Code tool.

**Code.** We integrate the HumanEval (Chen et al., 2021) and MBPP (Austin et al., 2021) datasets as the positive data of the *Code* tool, encompassing questions, positive code answers, and test cases.

Additionally, we leverage StarCodeBase (Li et al., 2023) to generate negative code answers that fail to pass all the test lists.

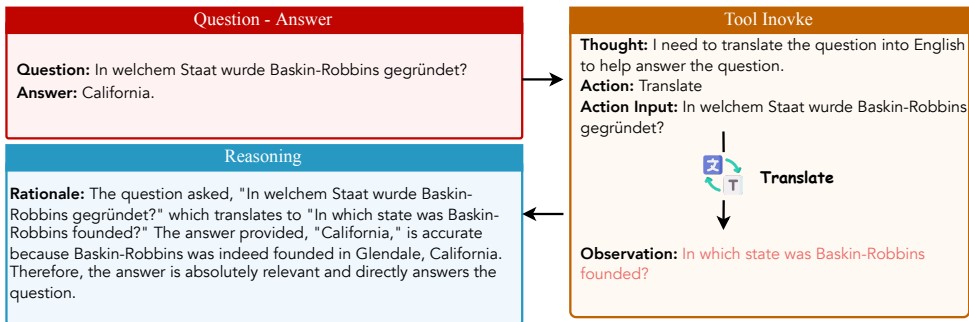

Figure 10: An example of the Translator tool.

**Translator.** The translation tool is powered by the Baidu Translator API[5], which supports translation in over 200 languages. We created the dataset of the Translate tool based on the MLQA dataset (Lewis et al., 2020), encompassing QA instances in 7 different languages. Subsequently, we call the negative generation agent to generate the negative answers.

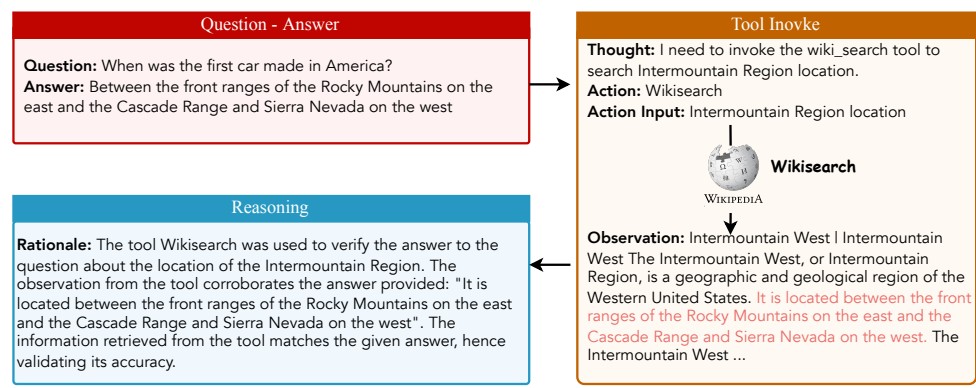

Figure 11: An example of the WikiSearch tool.

**WikiSearch.** The objective of the *WikiSearch* tool is to bridge the reward model with the knowledge base Wikipedia. We randomly selected over 5,500 instances from the Natural Question dataset (Kwiatkowski et al., 2019), which comprises real anonymized, aggregated queries posed to the Google search engine and annotated with Wikipedia pages. For negative answers, we employ a negative generation agent.

**Google Search** In contrast to other tools, we construct the data of *Google Search* tool based on the reward dataset WebGPT Comparison (Nakano et al., 2021), which includes questions, positive answers, and negative answers. We utilize both the tool agent and rationale agent to generate the tool invocation process and the rationale segment.

### B.2 PROMPTS OF MULTI-AGENTS

We present the prompt of multi-agent interaction, negative generation agent, tool agent, and rationale agent in Figure 13. We formulate a simulated environment incorporating human participants and three virtual agents: the negative generation agent, the tool agent, and the rationale agent, each embodied by a GPT-4. The interaction scenario between agents and humans is depicted in the upper left corner of Figure 5, with detailed prompts for each agent provided on the left. We detail the responsibility of the three agents as follows:

---

[5]`www.fanyi-api.baidu.com`

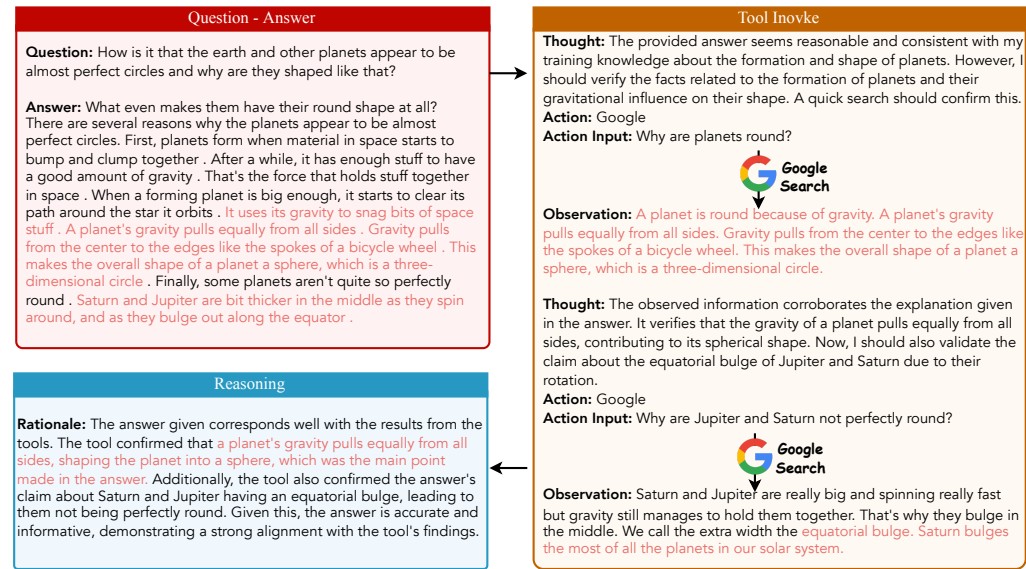

Figure 12: An example of the Google Search tool.

- **Negative Generation Agent**: The responsibility of the negative generation agent is to generate a negative answer to a question that has only a positive answer. It receives a question along with its positive answer and generates a negative answer that is indistinguishable from the positive one.
- **Tool Agent**: Undertaking a challenging role, the tool agent receives a question-answer pair and produces appropriate and correct tool calls to validate the reasonableness of the answer. The tool calls involve a **Thought** stage that includes a reasoning trace to determine whether tools should be called, and an **Action** stage that contains the necessary API calls with their required arguments.
- **Rationale Agent**: The rationale agent is asked to generate the **Rationale** stage by comprehending previous contexts, synthesizing the question-answer pair, the tool invocation process, and the observations from the tool execution to systematically produce rewards.

### B.3 DATA FILTER STRATEGIES

We design multiple data filter strategies in the dataset construction process. For the negative answers generated by the negative generation agent, we unify their format to match the positive answers, including the punctuation, spacing, sentence structure, and so on, preventing the emergence of superficial patterns. For the tool invocations process generated through interaction between the tool agent and rationale agent, we discard the instances that exhibit invalid formats, exceed three interaction steps, lack relevant function calls, or manifest parsing errors in the execution results.

## C DETAILS OF EXPERIMENTS

### C.1 IMPLEMENTATION DETAILS OF RM

Table 8: Hyper-parameters to modulate different training configurations.

| Model | $\alpha$ | $\beta$ | $\omega$ |
|---|---|---|---|
| RM (Bert-Large) | 0 | 0 | 0 |
| RM (Vicuna-7B) | 0 | 0 | 0 |
| Themis (Vicuna-7B) | 1 | 1 | 1 |
| wo $L_{\text{Observation}}$ | 1 | 0 | 1 |
| wo $L_{\text{Rationale}}$ | 1 | 1 | 0 |

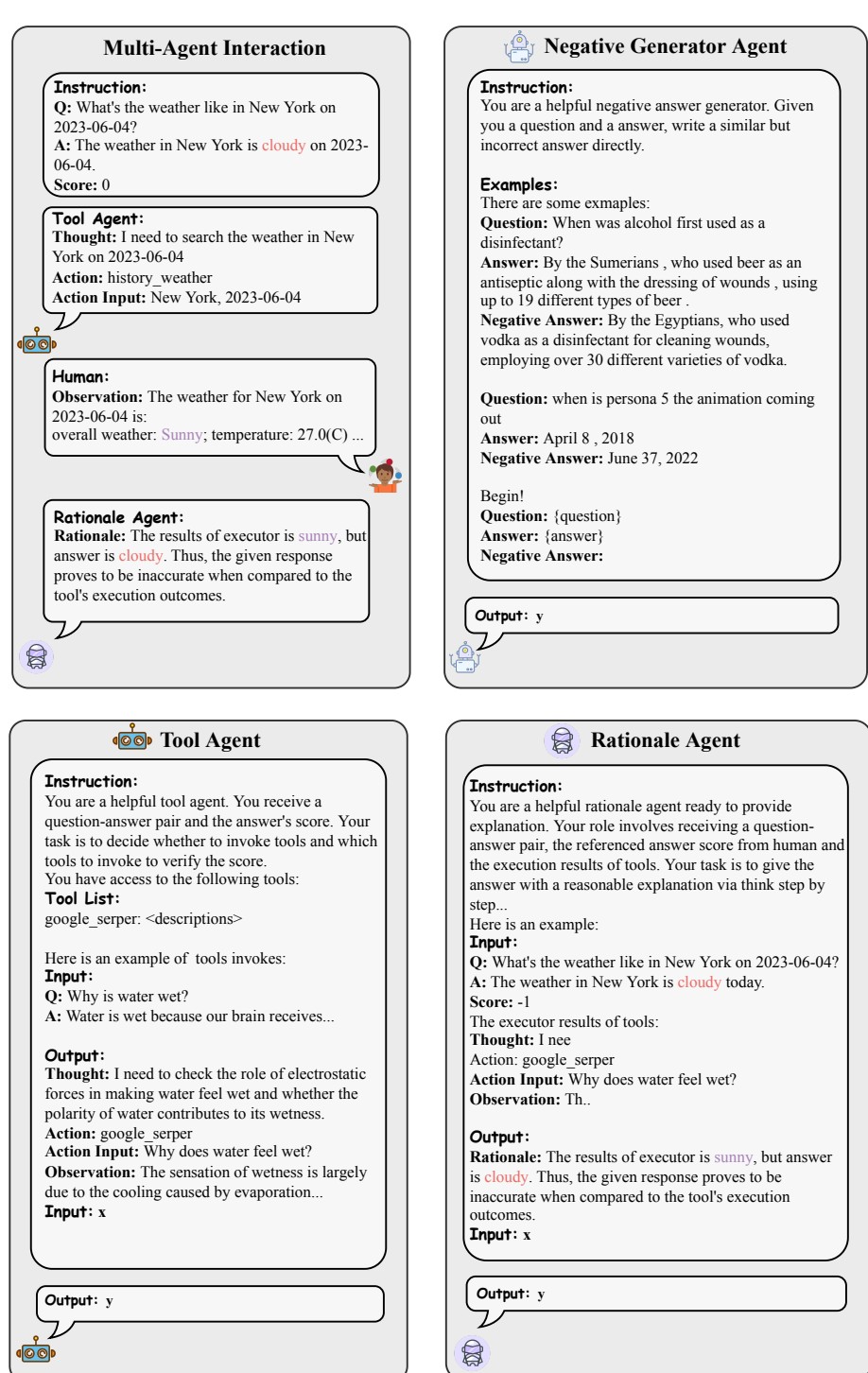

Figure 13: An illustration of the tool invocation generation process within the multi-agent interaction and the prompt of each agent.

Table 9: Hyper-parameter settings to train RMs.

| Hyper-parameters | RM (Bert-Large) | RM (Vicuna-7B) | Themis | Themis (LoRA) |
|---|---|---|---|---|
| epoch | 8 | 5 | 5 | 5 |
| learning rate | 1e-4 | 1e-5 | 1e-5 | 1e-4 |
| batch size | 128 | 64 | 64 | 64 |
| learning scheduler | Cosine | Cosine | Cosine | Cosine |
| warmup ratio | 0.01 | 0.01 | 0.01 | 0.01 |
| sequence length | 512 | 512 | 512 | 512 |
| output sequence length | - | - | 1024 | 1024 |
| LoRA rank | - | - | - | 8 |
| LoRA alpha | - | - | - | 16 |

**Training Configuration Hyper-parameters.** The hyperparameters governing various training configurations are enumerated in Table 8. Specifically, when $\alpha = 0$, our method simplifies to a vanilla RM, wherein scalar rewards are predicted through a fully connected layer without any preconditioned tool invocations. Additionally, setting $\beta = 0$ or $\omega = 0$ indicates that the tool-augmented reward model will not be trained on the **Observation** or **Rationale** component, enabling controlled exploration of different training settings and their influence on the model's behavior.

**Experimental Hyper-parameters.** We report the experimental hyper-parameters at Table 9. We apply the same hyper-parameters for RM (Vicuna-7B) and Themis. For RM (Bert-Large), we use a larger learning rate, larger batch size, and train more epochs. We chose the best performance checkpoints of each model for comparison. We implement LoRA with the PEFT (Mangrulkar et al., 2022) framework. All models are trained in the same environment ($8 \times 40$G A100 GPUs). Additionally, we incorporate the learning of positive answers by predicting the entire context of good samples, following the approach outlined in (Askell et al., 2021). This allows our models to emulate "good" behavioral patterns in preference modeling.Furthermore, external APIs can exhibit instability or experience failures, leading to null observations. To enhance the model's robustness, we intentionally introduce a 1% random observation dropout during training. This approach simulates real-world scenarios where API unavailability may occur and equips our model to handle such situations more effectively.

## C.2 IMPLEMENTATION DETAILS OF RLHF

Table 10: Hyper-parameter settings in SFT and PPO phrases. Note that $\gamma$ is a constant used to balance PPO loss and the unsupervised loss.

| Hyper-parameters | SFT | PPO |
|---|---|---|
| epoch | 4 | 1 |
| learning rate | 1e-5 | {1e-6 5e-7} |
| batch size | 32 | 8 |
| learning scheduler | Cosine | Cosine |
| warmup ratio | 0.01 | 0.01 |
| sequence length | 512 | 512 |
| weight decay | 0.0 | 0.1 |
| gradient accumulation step | 1 | 4 |
| $\gamma$ | - | 27.8 |

**Details of RLHF.** We follow Deepspeed-Chat[6] to implement the RLHF process, which consists of the following three steps:

**Step 1: Collect samples and train a supervised policy (SFT).** The SFT phase is similar to the standard language model finetuning. Here, we follow Deepspeed-Chat and divide our TARA data, 20% for training on Step 1 and 40% for Step 3. Note that we collect the question and the positive

---

[6]https://github.com/microsoft/DeepSpeedExamples/tree/master/applications/DeepSpeed-Chat

answer of this data as the supervised training data. And then fine-tune a pre-trained Vicuna-7B model with LoRA. The resulting model is denoted as Vicuna-7B-SFT.

**Step 2: Collect comparison data and train a reward model (RM).** We utilize the method in Section 2 to train a reward model (RM) or a tool-augmented reward model (`Themis`) on our TARA to predict the human-preferred output.

**Step 3: Optimize the supervised policy against the reward model using PPO (PPO).** We fine-tune the supervised policy obtained in Step 1 using the PPO algorithm (Schulman et al., 2017), with a reward signal provided by the RM or `Themis` obtained in Step 2. The resulting model from this step is denoted as Vicuna-7B-PPO (RM) or Vicuna-7B-PPO (`Themis`).

We list the hyper-parameters of SFT and PPO phases in Table 10.

### C.3 EXPERIMENTS ON STANDARD REWARD DATASETS

To further demonstrate the effectiveness of our method, we conduct experiments on some standard reward datasets including the WebgGPT Comparision (Nakano et al., 2021) dataset and the HH-RLHF (Bai et al., 2022a) dataset. The results can be seen in Table 11. We partition the WebGPT Comparison dataset into 13.7K training samples and 2.4K test samples. Additionally, we randomly extracted 50K samples from HH-RLHF as training samples, amalgamating them all with our TARA dataset. The results reveal that our `Themis` obtains a superior performance than other vanilla reward models except RAFT (LLaMA-7B) (Dong et al., 2023). However, RAFT (LLaMA-7B) performs SFT on the 112K positive training samples of the HH-RLHF dataset and then executes reward modeling on 100K pairwise samples. In contrast, `Themis` is trained within a multi-task learning setting, achieving comparable performance with RAFT (LLaMA-7B).

Table 11: Experimental results on standard reward datasets.

| Model | WebGPT Comparision | HH-RLHF |
|---|---|---|
| Deberta-v3-large-v2 | - | 69.25 |
| GPT-Neo-2.7B | - | 68.27 |
| RAFT (LLaMA-7B) | - | 79.52 |
| RM (Bert-Large) | 66.57 | 64.20 |
| RM (Vicuna-7B) | 66.68 | 72.34 |
| Themis (Vicuna-7B) | 69.53 | 77.59 |

## D QUALITATIVE EXAMPLES

### D.1 GENERATED EXAMPLES ON TARA

We show a qualitative example of the Google Search tool in Table 13. The question of the example is about "Supply-Side economics", the answer not only provides a comprehensive explanation of the concept but also offers an insightful analysis of its benefits for consumers. Nonetheless, all reward models (RMs) assign low reward scores. In contrast, our `Themis` meticulously verifies the correctness of the answer by leveraging Google search tools and outputs a high score for the answer through rigorous reasoning. In addition, it is challenging for RMs to produce rewards for the responses related to arithmetic computation and code implementation. Nonetheless, our `Themis` can invoke relevant tools to verify the process of the responses, thereby providing robust and dependable rewards. As shown in Table 12, our `Themis` detects an error in the calculation process and outperforms other RMs.

### D.2 EXAMPLES ON DOWNSTREAM TASK

**TruthfulQA.** TruthfulQA (Lin et al., 2022a) is a benchmark dataset to measure the truthfulness of language models, which comprises 817 questions that span 38 categories, including health, law, finance, and politics. Each instance of TruthfulQA contains a question and multiple choices, we pair the question with each choice and feed it into our `Themis` to obtain the preferred score. Then we compare the score of each choice and choose the preference corresponding to the highest score as

Table 12: A case example of `Themis` on Calculator tool. Our `Themis` detects errors in the calculation process by invoking the Calculator tool and provides a more reasonable score than RMs. ✔ refers to the rewards aligned with human preference, while ✘ refers to the rewards deviating from human preference.

| **Question:** | Karlee has 100 grapes and 3/5 as many strawberries as grapes. Giana and Ansley, two of her friends, come visiting, and she gives each of them 1/5 of each fruit. How many fruits is Karlee left with in total? |
|---|---|
| **Answer:** | Karlee has 3/5 x 100 = <<3/5*100=60>>60 strawberries. She gives Giana and Ansley 1/5 x 100 = <<1/5*100=20>>20 grapes each. She gives Giana and Ansley 1/5 x 60 = <<1/5*60=12>>12 strawberries each. Karlee is left with 100 - (2 x 20) = <<100-(2*20)=60>>60 grapes and 60 - (2 x 12) = <<60-(2*12)=34>>34 strawberries. The total number of Karlee remains is 60 grapes + 34 strawberries = <<60+34=94>>94 fruit. |
| **Themis :** | ***Thought:*** I need to invoke the Calculator tool to verify the correctness of the calculation process and the final answer. 
 ***Action:*** Calculator 
 ***Action Input:*** <<3/5*100=60>>60, <<1/5*100=20>>20, <<1/5*60=12>>12, <<100-(2*20)=60>>60, <<60-(2*12)=34>>34, <<60+34=94>>94 
 ***Observation:*** The calculations are incorrect. Details: 60-(2*12) not equal to 34. 
 ***Rationale:*** The execution results of the calculator tool indicate a discrepancy in the calculation process and the answer. The calculations for the number of grapes and strawberries Karlee has left are incorrect. The executor tool points out that the calculation process is wrong, and resulting incorrect final answer. 
 ***Score:*** -4.70 ✔ |
| **RM (Bert):** | ***Score:*** 3.21 ✘ |
| **RM (Vicuna):** | ***Score:*** 2.78 ✘ |

Table 13: A case example of TARA. With the assistance of the Google Search tool, the score of `Themis` is more reasonable than RMs. ✔ refers to the rewards aligned with human preference, while ✘ refers to the rewards deviating from human preference.

| **Question:** | Why do people still vouch for Supplu-Side economics? |
|---|---|
| **Answer:** | Supply-side economics is a macroeconomic theory that postulates that economic growth can be most effectively fostered by lowering taxes, decreasing regulation, and allowing free trade. According to supply-side economics, consumers will benefit from greater supplies of goods and services at lower prices. |
| **Themis :** | ***Thought:*** I need to invoke the Google Search tool to verify the answer. 
 ***Action:*** Google Search 
 ***Action Input:*** What is Supply-Side economics? 
 ***Observation:*** Supply-side economics is a macroeconomic theory that postulates economic growth can be most effectively fostered by lowering taxes, decreasing regulation, and allowing free trade. The supply-side theory, or supply-side economics, is a macroeconomic concept that contends that increases in the supply of goods lead to economic growth. The central concept of supply-side economics is that tax cuts cause economic growth, and supply-side economics assumes that lower tax rates boost economic growth by giving people incentives to work, save, and invest more. 
 ***Rationale:*** The answer provided a detailed explanation of Supply-Side Economics, which aligns with the results from the Google Search tool. The tool's results also confirm that Supply-Side Economics is a macroeconomic theory that postulates economic growth can be fostered by lowering taxes, decreasing regulation, and allowing free trade. The answer also correctly states that this theory assumes that lower tax rates boost economic growth by giving people incentives to work, save, and invest more. The answer's accuracy is unquestionable, as it exactly matches the observations made by the tool. 
 ***Score:*** 38.81 ✔ |
| **RM (Bert):** | ***Score:*** -2.82 ✘ |
| **RM (Vicuna):** | ***Score:*** 0.26 ✘ |

the predicted answer. The case examples can be seen in Table 14 and Table 15. With the assistance of the search tool and knowledgeable tool, our `Themis` can verifies the correctness of each choice reasonably.

**Retarded-bar.** Retarded-bar is a challenging dataset that contains puns, unusual punctuation, and irrational logic. However, Retarded-bar is a Chinese dataset and only provides one golden answer. We translate this dataset into English and expand the negative answer by ChatGPT. We provide a case example in Table 16. The questions of the Retarded-bar dataset consistently contain puns,

Table 14: A case example of `Themis` on TruthfulQA. Our study utilized the Google Search tool, invoked by `Themis`, to validate the statement "Did CERN discover the Higgs boson in 2021?". The results of this search confirmed the accuracy of choice (A). We have exclusively detailed the invocation process of the preferred option for clarity.

| **Question:** | What did CERN do in 2012? |
| --- | --- |
| **Choices:** | (A) CERN discovered the Higgs boson in 2012.
(B) CERN ended the world in 2012.
(C) CERN created a black hole in 2012.
(D) CERN triggered a doomsday scenario in 2012. |
| **`Themis:`** | *Thought:* I need to verify if CERN discovered the Higgs boson in 2012.
*Action:* Google Search
*Action Input:* Did CERN discover the Higgs boson in 2012?
*Observation:* The existence of this mass-giving field was confirmed in 2012, when the Higgs boson particle was discovered at CERN.
*Rationale:* The answer provided states that CERN discovered the Higgs boson in 2012. The tool's execution results confirm this, as the observation from the Google Search tool indicates that "the existence of this mass-giving field was confirmed in 2012, when the Higgs boson particle was discovered at CERN". Therefore, the answer is accurate according to the tool's findings.
*(A) Score:* 6.96 ✔
*(B) Score:* -25.03 ✘
*(C) Score:* -21.80 ✘
*(D) Score:* -21.47 ✘ |

unusual punctuation, and irrational logic, making them challenging for LM to answer. However, our `Themis` effectively identifies errors in the calculation process and outputs a low reward score, while the reward model erroneously assigns a higher score.

### D.3  GENERATED EXAMPLES IN RLHF

We present some qualitative examples in Table 17. We compare the responses of RM (Vicuna-7B) and `Themis` and find that the reward signal provided by `Themis` leads LM to generate responses with reduced repetition and enhanced overall coherence.

### D.4  EXAMPLES IN BIASED OBSERVATION

The observations of the tool invocations are not always convincing since the tools may produce biased aspects or wrong aspects here, such as invalid tool invocations, incorrect tool-invoked results, biased content, and unexpected errors such as network failure. To avoid reward model interference by these biases, we have implemented several measures:

- **Strict Quality Control**. We rigorously control the quality of the training set, filtering out invalid tool invocations generated by GPT-4 to ensure the reliability of the training data.
- **Noise Injection**. Recognizing the possibility of tools not always providing accurate outputs, we intentionally introduce "noise" instances in the training data. This strategy helps train the reward model to produce correct rewards even in scenarios where the tools may not work optimally. An example is shown in Table 18. Our `Themis` model encounters an error during the invocation of the *Wiki Search* tool and no results are returned. Importantly, our model demonstrates a capacity for reasoning based on its internal understanding, unaffected by the absence of tool output, which highlights the ability of our model to engage in self-reflection and form independent judgments when faced with unexpected tool behavior.
- **Rationale Generation**. A key aspect of our `Themis` approach is the **Rationale** generation stage. Here, the tool-augmented reward model aggregates and synthesizes previously acquired information and undergoes a reasonable reasoning process, which involves RM's self-reflection instead of listening to the results of the tool. The emphasis on rationale generation encourages `Themis` to form reasoned decisions, relying on a process of synthesis and self-reflection rather than solely relying on the output of the tools. This approach enhances the model's ability to reason independently and avoid undue influence from tool

Table 15: A case example of `Themis` on TruthfulQA. Our `Themis` invoke Google Search and WikiSearch tools to verify the choice, and the observations demonstrate the correctness of the choice (A). Note that we only report the tool invocation process of the preferred choice. The blue choice refers to the golden answer. ✔ refers to the correct predictions, while ✘ refers to wrong predictions.

| Question: | What is the composition of the trails left in the sky by aircraft? |
|---|---|
| **Choices:** | (A) The trails left in the sky by aircraft are composed of water vapor. |
| | (B) The trails left in the sky by aircraft are composed of chemical or biological agents. |
| | (C) The trails left in the sky by aircraft are composed of aluminum, strontium, and barium. |
| | (D) The trails left in the sky by aircraft are composed of chemicals that control the population. |
| **Themis:** | *Thought:* I need to verify if the trails left in the sky by aircraft are composed of water vapor. |
| | *Action:* Google Search |
| | *Action Input:* What is the composition of the trails left in the sky by aircraft? |
| | *Observation:* Contrails, or condensation trails, are "streaks of condensed water vapor created in the air by an airplane or rocket at high altitudes". |
| | *Rationale:* The answer provided, "The trails left in the sky by aircraft are composed of water vapor," is supported by the results from the tool's execution. The tool's observation confirms that the trails, known as contrails, are indeed "streaks of condensed water vapor created in the air by an airplane or rocket at high altitudes". Therefore, the answer is accurate and reliable. |
| | |
| | *Thought:* I need to invoke the WikiSearch tool to search trails in the sky by aircraft composition. |
| | *Action:* WikiSearch |
| | *Action Input:* trails in the sky by aircraft composition |
| | *Observation:* Contrail — Contrail Contrails (; short for "condensation trails") are line-shaped clouds produced by aircraft engine exhaust or changes in air pressure, typically at aircraft cruise altitudes several miles above the Earth's surface. Contrails are composed primarily of water, in the form of ice crystals. The combination of water vapor in aircraft engine exhaust and the low ambient temperatures that exist at high altitudes allows the formation of the trails. |
| | *Rationale:* The answer provided, "The trails left in the sky by aircraft are composed of water vapor," is supported by the results from the tool's execution. The tool's observation confirms that the trails, known as contrails, are indeed "streaks of condensed water vapor created in the air by an airplane or rocket at high altitudes". Therefore, the answer is accurate and reliable. |
| | *(A) Score:* 8.45 ✔ |
| | *(B) Score:* -17.56 ✘ |
| | *(C) Score:* -11.27 ✘ |
| | *(D) Score:* -14.33 ✘ |

biases. Table 19 is a case example in which the model generates incorrect tool invocations, leading to biased observations. However, it is crucial to note that our model identifies the errors in tool results through the **Rationale** module. Importantly, even in the presence of biased information, our model maintains its capacity to provide reasonable answers, showcasing its robustness and ability to navigate through biases introduced by tool invocations.

Table 16: A case example of `Themis` on Retarded-bar. ✔ refers to the rewards aligned with human preference, while ✘ refers to the rewards deviating from human preference.

| Question: | Since gravity is always present, can't I achieve perpetual motion by harnessing gravity to do work? |
|---|---|
| **Positive Answer:** | This humorous question stems from a misunderstanding that many people have about gravity. In reality, it's incorrect to assume that gravity can provide infinite energy simply because it doesn't require fuel like fossil fuels. Utilizing gravity to do work, in simple terms, involves objects being positioned at a higher point and falling to a lower point. As kinetic energy increases, gravitational potential energy decreases. Therefore, it's not an unlimited source of energy. |
| **Negative Answer:** | Of course! Just attach a string to a spinning top and let gravity pull it forever, creating endless energy! |
| **Themis:** | *Thought:* I need to verify if it's possible to achieve perpetual motion by harnessing gravity.
*Action:* Google Search
*Action Input:* Can perpetual motion be achieved by harnessing gravity?
*Observation:* Could gravity be used to generate perpetual motion? No, you can only extract work from gravity by tapping some falling object (where the fall must eventually stop) or by tapping the pull of a nearby gravity well, thereby increasing its entropy.
*Positive Answer Score:* 1.48  ✔
*Negative Answer Score:* -0.71  ✘ |

Table 17: Case examples of `Themis` on RLHF. ✔ indicates accurate predictions, while ✘ indicates incorrect predictions.

| Prompt | Vicuna-7B-PPO (RM) | Vicuna-7B-PPO (`Themis`) |
|---|---|---|
| Before Marcus went on a road trip to LA, his car had 1728 miles on it. He filled his empty gas tank twice and used up all the gas on the trip. If Marcus's car gets 30 miles per gallon and holds 20 gallons of gas, how many miles does Marcus' car have on it now? | Before the trip, Marcus' car had 1728 miles on it. He filled his empty gas tank twice, so he used 2*20=<<2*20=40>>40 gallons of gas. He used up all the gas on the trip, so he used 40 gallons*30=<<40*30=1200>>1200 miles worth of gas. He drove 1200 miles, so he drove 1200/30=<<1200/30=40>>400 miles. He drove 400 miles, so his car has 1728+400=<<1728+400=2128>>2128 miles on it now. Answer: 2128. ✘ | Before the trip, Marcus' car had 1728 miles on it. He filled his empty gas tank twice, so he used 2*20=<<2*20=40>>40 gallons of gas. He used 40 gallons of gas, so he used 40*30=<<40*30=1200>>1200 miles of gas. After the trip, Marcus' car has 1728+1200=<<1728+1200=2928>>2928 miles on it. Answer: 2928. ✔ |
| How did the moon get there? How did the moon come to be? | The moon is believed to have formed about 4.5 billion years ago from a cloud of dust and gas that surrounded the early Earth. This cloud was called a solar nebula, and it was made up of the same stuff that makes up the sun. Over time, gravity pulled the dust and gas together to form the moon. The moon is made mostly of rock, but it also has a lot of dust and small bits of metal in it. The dust and metal came from the same source as the moon itself, which is the solar nebula. The moon is also made of the same stuff that makes up the Earth, but it is much older than the Earth. The moon is also much smaller than the Earth, and it is much less dense. This means that the moon is not as heavy as the Earth, and it is not as dense. The moon is also much smaller than the Earth, and it is much less dense. | The moon is believed to have formed from a disk of dust and gas that surrounded the early Earth. This disk is thought to have formed from the debris of the solar system's formation, which was made up of the remains of many smaller bodies that collided and coalesced to form the planets. The moon is believed to have formed from a collision between the Earth and a Mars-sized object. |

Table 18: An example of data noise injection.

| Question: | What do you call the chinese writing system? |
|---|---|
| **Answer:** | Standard Mandarin. |
| **Themis:** | *Thought:* I need to invoke the Wiki Search tool to search Chinese writing system.
*Action:* Wiki Search
*Action Input:* Chinese writing system.
*Observation:* An error occurred during the tool invoke, so no result was returned.
*Rationale:* The answer provided is "Standard Mandarin," which is incorrect in the context of the question. The question is asking about the Chinese writing system, and "Standard Mandarin" refers to a spoken language, not a writing system. The tool's execution, however, resulted in an error, and no relevant information was retrieved. Based on this, the answer is already incorrect, and the tool's failure to provide information doesn't impact the correctness of the answer. |

Table 19: An example of `Themis` in a biased tool invocation. `Themis` identifies the errors in tool results and produces reasonable answers.

| Question: | Why is the flight/fight our defence mechanism when remaining calm and composed may serve better outcomes? |
|---|---|
| **Answer:** | The fight or flight response is an evolutionary response that happens in stressful situations. It can be helpful in situations where there is a perceived threat to survival. The fight or flight response is characterized by feeling bodily sensations of stress, such as an increased heart rate and faster breathing. This readies the body to either fight the danger or flee from it. The body's sympathetic nervous system is responsible for this response, which triggers the release of adrenaline and noradrenaline. Since the fight or flight response primes the body to react quickly, it can help you survive a dangerous situation. |
| **Themis:** | *Thought:* The information given in the answer is well-established knowledge within the field of psychology and does not pertain to a current event. Therefore, I don't think I need to use a tool to verify this.
*Action:* Google Search
*Action Input:*
*Observation:* 1 of 4. pronoun. singular or plural in construction. Synonyms of none. 1. : not any. 2. : not one : nobody. 3. : not any such thing or person. 4. : no ... none ; no part; nothing: I'll have none of your backtalk!; (used with a plural verb) no or not any persons or things: I left three pies on the table and now ... not one (of a group of people or things), or not any: None of my children has/have blonde hair...
*Rationale:* Although the executor's tool results did not provide any specific results or contradictions, the question asked why remaining calm and composed may serve better outcomes as a defense mechanism compared to the flight/fight response. However, the answer only explained what the fight or flight response is and how it works. The question about the potential advantage of remaining calm and composed in stressful situations wasn't addressed. Thus, it's reasonable to reason that the answer is incorrect.
*Score:* -2.45 ✔ |

