# OpenReview forum: "Tool-Augmented Reward Modeling"
_ICLR.cc/2024/Conference — ICLR 2024 spotlight_

### Official Review · Reviewer_zDVk · 2023-10-29

**Soundness:** 2 fair
**Presentation:** 2 fair
**Contribution:** 4 excellent
**Rating:** 8
**Confidence:** 4

**Summary:**

This paper investigates how to augment reward models with tools and proposes a new training framework, THEMIS, to do so. The Themis approach trains the model to select the correct tool for a given prompt and completion and reason about how the tools used should impact the assigned preference. In addition, the authors provide a tool-related RM dataset.

**Strengths:**

**Strength 1**: The idea of augmenting reward models with tools is very interesting, novel, and timely.

**Strength 2**: The proposed method provides a nice and logical way for tools to be included in the reward design process.

**Strength 3**: This paper provides some interesting experiments such as application to RLHF and scaling experiments.

**Weaknesses:**

**Weakness 1**: One of my main concerns is lack of experiments on standard reward modeling datasets. There are many datasets not included in the paper such as the Anthropic HH dataset, Stack Overflow, OpenAI WebGPT, and ChatGPT comparisons datasets. They do conduct analysis on a small portion of the HH dataset, but not on the provided testing set. In addition, they show worst test accuracy than is reported in some other papers that only use conventional reward modeling [1]. Since the main claim of the paper is that by using tools they can improve the accuracy of reward models, I think their method should be validated on these popular datasets.

[1] Dong, Hanze, et al. "Raft: Reward ranked finetuning for generative foundation model alignment." arXiv preprint arXiv:2304.06767 (2023).

**Weakness 2**: Little Hyperparameter study. THEMIS introduces various hyperparameters, but the sensitivity of model performance to these hyperparameters is not discussed.

**Weakness 3**: This paper does not discuss a significant limitation of this method: the difficulty of creating the dataset. The dataset creation process consists of various complex steps, involves tool selection and design of heuristics. This seems to be difficult to scale to large scale preference datasets.


**Weakness 4**: I think that this paper could use a more in depth discussion of related works. In particular, various works have attempted to use similar tools such as a compiler in the reward design process [2,3,4,5] and [4] use it to guide the reward model training. Discussing these works could help better frame the contribution of this work.

[2] Le, Hung, et al. "Coderl: Mastering code generation through pretrained models and deep reinforcement learning." Advances in Neural Information Processing Systems 35 (2022): 21314-21328.

[3] Shen, Bo, et al. "Pangu-coder2: Boosting large language models for code with ranking feedback." arXiv preprint arXiv:2307.14936 (2023).

[4] Bukharin, Alexander, et al. "Deep Reinforcement Learning from Hierarchical Weak Preference Feedback." arXiv preprint arXiv:2309.02632 (2023).

[5] Shojaee, Parshin, et al. "Execution-based code generation using deep reinforcement learning." arXiv preprint arXiv:2301.13816 (2023).

**Questions:**

How exactly do you get the scalar reward from the reward model? Is this done with a separate output layer?

---

> ### Author Response · Authors · 2023-11-19
> **Response to Reviewer zDVk**
>
> Dear Reviewer zDVk,
>
> Many thanks for your detailed and constructive comments.
>
> **Response to weaknesses**:
>
> **For W1**: Thank you for your valuable feedback. In response to your concern, we have conducted experiments on two standard reward datasets: WebGPT Comparison and HH-RLHF. The results are as follows: We partitioned the WebGPT Comparison dataset into 13.6K training samples and 2.4K test samples. Additionally, we randomly extracted 50K samples from HH-RLHF as training samples, combining them with our TARA dataset. The results reveal that our Themis outperforms other vanilla reward models, with the exception of RAFT (LLaMA-7B). It's noteworthy that RAFT (LLaMA-7B) performs SFT on the 112K positive training samples of the HH-RLHF dataset and then executes reward modeling training on 100K pairwise samples.
> In contrast, Themis does not perform SFT and is exclusively trained on 50K samples from HH-RLHF (vs. RAFT trained on 100k samples on RMs as well as SFT on 112K samples) in a multi-task learning setting (WebGPT Comparison + HH-RLHF + TARA), achieving comparable performance with RAFT (LLaMA-7B). We believe these additional experiments address your concerns and provide valuable insights into the effectiveness of our approach on standard reward modeling datasets. Thank you for bringing this to our attention, and we appreciate your thoughtful review.
>
> | Model | WebGPT Comparisons | HH-RLHF |
> | --- | --- | --- |
> | Deberta [1] | - | 69.25 |
> | GPT-Neo-2.7B [2] | - | 68.27 |
> | RAFT (LLaMA-7B) [2] | - | **79.52** |
> | RM (Bert-Large) | 66.57 | 64.20 |
> | RM (Vicuna-7B) | 66.68 | 72.34 |
> | Themis (Vicuna-7B) | **69.53** | 77.59 |
>
> **For W2**: There are two types of hyper-parameters in our work: training configurations ($\alpha, \beta, \omega \in \{0, 1\}$ in Sec 2.2) and experiment hyper-parameters (such as batch size and learning rate). The hyper-parameters $\alpha, \beta, \omega$ correspond to different training settings as shown in the following:
> We have provided a comprehensive list of experiment hyperparameters and training details in Appendix C.1, specifically in Table 8 and Table 9 of the submitted version. However, it's crucial to acknowledge that we did not conduct an extensive hyperparameter tuning (especially for batch sizes) in this study due to limitations in our training resources. To maximize resource utilization, we opted for the largest feasible batch size and set the learning rate to 1e-5. We appreciate your observation and acknowledge the potential value of a more extensive hyperparameter study, which could be explored in future work with additional resources.
>
> | Model | $\alpha$ | $\beta$ | $\omega$ |
> | --- | --- | --- | --- |
> | RM (Bert-Large) | 0 | 0 | 0 |
> | RM (Vicuna-7B) | 0 | 0 | 0 |
> | Themis | 1 | 1 | 1 |
> | &nbsp;&nbsp;&nbsp; w/o $L_\text{Observation}$ | 1 | 0 | 1 |
> | &nbsp;&nbsp;&nbsp; w/o $L_\text{Rationale}$ | 1 | 1 | 0 |
>
> **For W3**: Thank you for your insightful comment. We appreciate your attention to the dataset creation challenges, and we have incorporated a discussion of these limitations in the Limitation section of our paper. Despite the inherent complexities, we have implemented measures to streamline the process and facilitate scaling to larger preference datasets.
> To address scalability concerns, we have established an automatic pipeline for dataset construction involving multi-agent interactions using LLMs. Furthermore, we have integrated standard preference data (hh-rlhf) with our tool-augmented data during reward model training. The resulting model exhibits superior generalization performance compared to the vanilla reward model, as detailed in Section 4.4.
> While tool selection and heuristic design are integral to tool-augmented datasets, we have automated and accelerated these processes for scalability. By defining the tool invoke format, we can generate numerous instances through our automatic pipeline, contributing to efficient scaling. We hope this can answer your question.
> Furthermore, we will open-source our tool-augmented dataset and code to the research community. We believe this transparency will foster collaboration and further development in this area. Thank you for your valuable feedback, and we look forward to contributing to the broader research community.
>
> **For W4**:
> We sincerely appreciate your valuable feedback. In response to your suggestion, we have enhanced the discussion on related works in Section 5 of the paper. The updated section now delves into the works you referenced, providing a more comprehensive context that helps better frame the contribution of our work in relation to existing literature. We hope this revision meets your expectations, and we thank you for bringing this to our attention.
>
> We trust that the aforementioned clarification could address your concerns. We genuinely look forward to engaging in further discussion and appreciate your time and consideration. Thank you for valuable feedback!

---

> ### Author Response · Authors · 2023-11-19
> **References**
>
> [1] https://huggingface.co/OpenAssistant/reward-model-deberta-v3-large-v2
>
> [2] Dong, Hanze, et al. "Raft: Reward ranked finetuning for generative foundation model alignment." arXiv preprint arXiv:2304.06767 (2023).

---

> > ### Comment · Reviewer_zDVk · 2023-11-21
> > **Thank you for the detailed response.**
> >
> > Thank you for taking the time to run additional experiments and answering my questions. My concerns have largely been addressed.
> >
> > I just have one follow up question. For the WebGPT and HH datasets, is the reason for Themis's superior performance due to the use of the search tool?

---

> ### Author Response · Authors · 2023-11-22
> **Response to Reviewer zDVk Follow-Up**
>
> Thank you for your feedback and for revisiting our experiments! The superior performance of Themis on the WebGPT and HH datasets can be attributed to two key factors:
> 1. **Utilization of Search Tools**:
>     * Themis leverages the Wiki Search and Google Search tools, which prove highly beneficial for tasks associated with the WebGPT and HH-RLHF datasets. By tapping into these search tools, Themis gains access to additional pertinent information, enhancing its ability to comprehend and address the intricacies of the tasks.
> 2. **Benefit of the Rationale Stage**:
>     * The incorporation of the Rationale stage in Themis prompts a Chain-of-Thought (CoT) approach to reward generation. During this stage, the tool-augmented reward model engages in information aggregation, synthesis, and self-reflection. This process empowers Themis to navigate through complex scenarios, providing a nuanced and contextually relevant basis for generating rewards. The Rationale stage plays a pivotal role in Themis's capacity to reason effectively and contribute to its superior performance on these datasets.
>
> We value your continued engagement and hope that this response could adequately addresses your concerns. If you have any further questions or require additional clarification, please feel free to reach out.

---

### Official Review · Reviewer_wTHw · 2023-10-30

**Soundness:** 3 good
**Presentation:** 2 fair
**Contribution:** 3 good
**Rating:** 8
**Confidence:** 3

**Summary:**

This paper suggests leveraging primitive tools in the reward model to enhance reward estimation for RLHF settings. The authors conducted extensive experiments to show that by having tool-grounded reward models, it is possible to boost the reward model accuracy and translate them to RLHF gains.

Use of tools such as web search or calculator to improve the performance of an LLM is not particularly new. I understand that the current literature has mostly focused on inherent knowledge in the models at the alignment training.

** updated score after reading the response and revisions

**Strengths:**

- Having a level of reasoning and interpretability is great feature to have for reward models
- The experiments and provided implementation details look comprehensive

**Weaknesses:**

1- How to trust tools is an important aspect to consider here. At least in the examples, it looks like there is a risk of biasing the reward model and generative model to outputs of specific tools being used. This could be concerning as tools are not necessarily unbiased.

2- It is not entirely clear how GPT-4 is used to generate RM training data. Note that GPT-4 itself is a system if the proposal is to use GPT-4 to train RM, one can argue why not directly train RM on GPT-4 data or use GPT-4 directly as reward model.

3- When alpha in eq. 2 is set to zero, we converge to typical tool use via simple prompting right? the RM is still a pretrained model and can be prompted to use tools even without explicit loss terms on tool use. Is this understanding correct?

4- Model size of 7B is quite small to capture knowledge to compete with the tools used in this paper. I think a more realistic setup would be to take a larger model and the gap between say Wiki tool and that result could look very different.

5- I think the write-up could improve, especially for Sec 3.1, I had difficulty understanding exactly how GPT-4 was used and how training data was prepared

**Questions:**

(see above)

**Details Of Ethics Concerns:**

No particular concern. Added one related point in my review, looking forward to hearing from authors.

---

> ### Author Response · Authors · 2023-11-19
> **Response to Reviewer wTHw (Part 1/3)**
>
> Dear reviewer wTHw,
>
> Many thanks for your detailed and constructive comments.
>
> **Response to weaknesses**:
>
> **For W1**. Your concern about the potential bias introduced by the tools is indeed crucial.  As you said, the tools may produce biased aspects or wrong aspects here, such as invalid tool invocations, incorrect tool-invoked results, biased content, and unexpected errors such as network failure. To avoid the reward model interference by these biases, we have implemented several measures:
> 1. **Strict Quality Control**: We rigorously control the quality of the training set, filtering out invalid tool invocations generated by GPT-4 to ensure the reliability of the training data.
> 2. **Noise Injection**: Recognizing the possibility of tools not always providing accurate outputs, we intentionally introduce "noise" instances in the training data. This strategy helps train the reward model to produce correct rewards even in scenarios where the tools may not work optimally. An example is shown in the following. Our Themis model encounters an error during the invocation of the *Wiki Search tool* and no results are returned. Importantly, our model demonstrates a capacity for reasoning based on its internal understanding, unaffected by the absence of tool output, which highlights the ability of our model to engage in self-reflection and form independent judgments when faced with unexpected tool behavior.
>
>     > Question: What do you call the chinese writing system?
>     >
>     > Answer: Standard Mandarin
>     >
>     > **Thought**: I need to invoke the Wiki Search tool to search Chinese writing system
>     >
>     > **Action**: Wiki Search
>     >
>     > **Action Input**: Chinese writing system
>     >
>     > **Observation**: An error occurred during the tool invoke, so no result was returned.
>     >
>     > **Rationale**: The answer provided is "Standard Mandarin," which is incorrect in the context of the question. The question is asking about the Chinese writing system, and "Standard Mandarin" refers to a spoken language, not a writing system. The tool's execution, however, resulted in an error, and no relevant information was retrieved. Based on this, the answer is already incorrect, and the tool's failure to provide information doesn't impact the correctness of the answer.
>
> 3. **Rationale Generation**: A key aspect of our Themis approach is the Rationale generation stage. Here, the tool-augmented reward model aggregates and synthesizes previously acquired information and undergoes a reasonable reasoning process, which involves RM's self-reflection instead of listening to the results of the tool.  The emphasis on rationale generation encourages Themis to form reasoned decisions, relying on a process of synthesis and self-reflection rather than solely relying on the output of the tools. This approach enhances the model's ability to reason independently and avoid undue influence from tool biases. This is an example in which the model generates incorrect tool invocations, leading to biased observations. However, it is crucial to note that our model identifies the errors in tool results through the Rationale module. Importantly, even in the presence of biased information, our model maintains its capacity to provide reasonable answers, showcasing its robustness and ability to navigate through biases introduced by tool invocations.
>
>     > Question: Why is the flight/fight our defence mechanism when remaining calm and composed may serve better outcomes?
>     >
>     > Answer: The fight or flight response is an evolutionary response that happens in stressful situations. It can be helpful in situations where there is a perceived threat to survival. The fight or flight response is characterized by feeling bodily sensations of stress, such as an increased heart rate and faster breathing. This readies the body to either fight the danger or flee from it. The body’s sympathetic nervous system is responsible for this response, which triggers the release of adrenaline and noradrenaline...
>     >
>     > **Thought**: The information given in the answer is well-established knowledge within the field of psychology and does not pertain to a current event. Therefore, I don't think I need to use a tool to verify this.
>     >
>     > **Action**: Google Search
>     >
>     > **Action Input**: None
>     >
>     > **Observation**: 1 of 4. pronoun · ˈnən. singular or plural in construction. Synonyms of none. 1. : not any. 2. : not one : nobody. 3...
>     >
>     > **Rationale**: Although the executor's tool results did not provide any specific results or contradictions, the question asked why remaining calm and composed may serve better outcomes as a defense mechanism compared to the flight/fight response. However, the answer only explained what the fight or flight response is and how it works. The question about the potential advantage of remaining calm and composed in stressful situations wasn't addressed. Thus, it's reasonable to reason that the answer is incorrect.

---

> ### Author Response · Authors · 2023-11-19
> **Response to Reviewer wTHw (Part 2/3)**
>
> **For W2**. GPT-4 is a closed-source model that poses significant challenges in terms of cost and accessibility. Additionally, the practicality of conducting reward model training on GPT-4 is hindered by the necessity to add extra fully connected layers for reward prediction since OpenAI API did not support this feature. Given its high cost and limited availability for public use, employing GPT-4 directly as a reward model becomes impractical for wider applications. Therefore, we just utilize the strong in-context learning ability of GPT-4 and prompt GPT-4 to generate partial data, including tool-invoking processes (*Thought* and *Action* in Sec 2.2) and rationales (*Rationale*). Subsequently, we meticulously filter out invalid tool invocations and incorrect rationales to ensure the quality and reliability of the generated training data.
>
> **For W3**.  Your question is insightful, and we appreciate the opportunity to clarify. When alpha is set to zero, the model transforms into a vanilla reward model, specifically corresponding to RM (Vicuna-7B/Bert-Large) as indicated in Table 1 and Equation 1. In this configuration, the model generates a scalar reward solely through a fully connected layer, devoid of any tool-invoking or explanation-generating processes.
> With alpha set to zero, it's crucial to note that the reward model cannot spontaneously produce a tool use process without explicit loss terms on tool use. While we can resort to in-context learning (ICL) by prompting the model with a few tool-use examples, it is important to highlight that this approach may still lag behind in finetuning on different tools (with $\alpha=1$). Additionally, it is constrained by the maximum context length, as the contextual examples of ICL can consume significant length.
>
> **For W4**. In Sec 4.2 (Scaling Trends in Themis), we explored the correlation between the performance and the scale of models. To enhance the training efficiency and reduce the resource cost, we utilized the parameter-efficient fine-tuning method LoRA which only trains specific modules, as shown in Table 1. The results elucidated a positive correlation between the scale of the model and its overall performance, and highlighted a substantial gap in the *WikiSearch* tool, as you rightly pointed out.
> Taking your valuable advice into consideration, we have conducted full-parameter fine-tuning with Themis (Vicuna-33B), and reported the results as follows.  As illustrated in the table below, Themis with 33B demonstrates superior performance compared to its 7B counterparts, exhibiting an approximately 2% absolute improvement in preference prediction. Notably, the larger models showcase optimal capabilities in harnessing *Wiki Search*, *Google Search*, and *Translator* tools. This trend aligns with the findings from our LoRA experiments, which we have incorporated into Table 1. We value your thoughtful review and trust that these additional experimental results adequately address your concerns.
> | Model | Calendar | Calculator | Weather | Code | Translator | WikiSearch | Google Search | Multi | Avg. |
> | --- | --- | --- | --- | --- | --- | --- | --- | --- | --- |
> | Themis (Vicuna-7B) | 100.00 | 98.05 | 100.00 | 99.47 | 90.91 | 93.31 | 64.92 | 99.31 | 93.31 |
> | Themis (Vicuna-33B) | 100.00 | 97.40 | 100.00 | 99.47 | 93.54 | 96.55 | 73.72 | 99.31 | 95.21 |

---

> ### Author Response · Authors · 2023-11-19
> **Response to Reviewer wTHw (Part 3/3)**
>
> **For W5**. Sorry for the confusion about the data collection steps. Due to the page limits, we have revised Section 3.1 and added detailed prompts for each agent in Appendix B in the submitted version for a more comprehensive understanding. Furthermore, we have also revised Section 3.1 in the draft version, hope this can address some of your concerns. We simplify the dataset construction steps as follows:
> * Step 1: Initially, we collect a reward dataset featuring each instance comprising a question, a positive answer, and a negative answer. To construct this dataset, we employ two distinct approaches: resume open-source, high-quality datasets, and generation from scratch with some heuristic methods. However, the above methods usually only produce positive answers. To address this concern, we leverage a GPT-4 as a negative generation agent to generate antagonistic negative answers, which will be described in Step 3.
> * Step 2: Subsequently, we define the name, and API calls of some tools including *Calculator, Code Interpreter, Translator, Google, Weather, Calendar Search*, and *Wiki Search*.
> * Step 3: In this step, we leverage the capabilities of  GPT-4 to generate the tool invocation processes and negative samples. To achieve this, we design a simulated environment where GPT-4 plays three different agents: *the negative generation agent*, *the tool agent*, and *the rationale agent*. Taking the Calculator tool as an example, we extract some instances from GSM-8K. However, the data only contain questions and positive answers. To address this limitation, we prompt the negative generation agent to generate the negative answer, constructing the pairwise preference samples. Subsequently, we query the tool agent to generate the tool invocation process, encompassing the *Thought*, *Action*, and *Action Input* stages. Then humans execute the tools based on the generated tool information, yielding the *Observation*. Finally, we amalgamate the above information (inputted question-answer, tool invocation process, observations) and prompt the rationale agent to produce the *Rationale* stage. For a more comprehensive understanding of this process, additional details are available in Appendix B.2.
> * Step 4: We exclude tool-invoked processes acquired in Step 3 that exhibit invalid formats. Subsequently, we discard processes exceeding three interaction steps, lacking relevant function calls, or manifesting parsing errors in their execution results.
>
> We hope the above can address some of your concerns.  And we sincerely look forward to further discussion. Thank you for your time and consideration!

---

> > ### Comment · Reviewer_wTHw · 2023-11-21
> > **Re: Response to Reviewer wTHw**
> >
> > Thank you for the detailed responses and revisions. Based on the responses and new information presented in the paper, I'm updating the score.

---

> ### Author Response · Authors · 2023-11-22
> **Response to Reviewer wTHw Follow-Up**
>
> We sincerely appreciate your thorough review and the time you took to consider our responses and revisions! We appreciate the reviewer's insightful feedback, which contributes to the robustness and clarity of our research.

---

### Official Review · Reviewer_bKyt · 2023-10-31

**Soundness:** 3 good
**Presentation:** 3 good
**Contribution:** 3 good
**Rating:** 6
**Confidence:** 4

**Summary:**

The paper proposes a tool-augmented preference modeling approach, named Themis, to address the limitations of conventional reward models (RMs) in aligning language models with human preferences. The approach integrates external tools into RMs, enabling them to interact with diverse external sources and construct task-specific tool engagement and reasoning traces. The paper presents experimental results demonstrating the effectiveness of Themis in enhancing the performance of RMs across various tasks. The authors also provide a comprehensive tool-augmented reward dataset (TARA) for further research.

**Strengths:**

(1) The paper addresses an important issue in reward modeling by introducing a tool-augmented approach to enhance the effectiveness of RMs.

(2) The proposed methodology of integrating external tools into RMs is innovative and practical, allowing for dynamic decision-making and reasoning processes.

(3) The experimental results demonstrate significant improvements in preference ranking and outperformance of Themis compared to baseline RMs, validating the effectiveness of the approach.

**Weaknesses:**

The description of the method is not very clear. My understanding is that the reward model first generates some explanations based on the inputted question and answer, and then connects a fully connected layer to the final hidden state to produce a scalar reward.

**Questions:**

1. Why is there a significant difference between the results of Themis (Vicuna-7B + LoRA) and Themis in table1?
2. Does beta=0, w=0 in table1 mean RM (Vicuna-7B)?
3. I understand that all the values in table2 are accuracies, so why is the binary classification accuracy lower than 50 for RM (Vicuna-7B, Zero-shot)?

---

> ### Author Response · Authors · 2023-11-19
> **Response to Reviewer bKyt**
>
> Dear Reviewer bKyt,
>
> Many thanks for your detailed and constructive comments.
>
> **Response to Weaknesses**:
>
> You accurately captured the essence of our approach. Our reward model initiates tool-invoking processes and generates explanations based on the inputted question and answer. Subsequently, we append an ``<eos>`` token, and the scalar reward is produced by a fully connected layer using the last hidden states corresponding to the ``<eos>`` token. To enhance clarity, we have revised the method description in Section 3, providing more details. We trust that these modifications improve the overall clarity of our method description.
>
> **Response to Questions**:
>
> **For Q1**: In Themis (Vicuna-7B + LoRA), we adopt a strategy where specific parameter-efficient modules (using **LoRA**) are trained, while the **pretrained weights of Vicuna are frozen**. This approach significantly reduces training resource costs but may lead to a performance decrease compared to the full-parameter fine-tuning setting.
>
> **For Q2**: RM (Vicuna-7B) represents a **vanilla reward model**. In this configuration, the model generates a **scalar reward solely through a fully connected layer**, without engaging in any tool-invoking or explanation-generating processes. Therefore, in the context of our methodology, RM (Vicuna-7B) aligns with the setting where $α=0$.
>
> **For Q3**: The observed lower performance in the **zero-shot setting** is primarily attributed to the fact that it was evaluated **without training on HH-RLHF datasets**. In contrast, the **fine-tuning** results demonstrate performance **exceeding 50%**. More specifically, the lower accuracy for RM (Vicuna-7B, Zero-shot) is a result of the specific design choices made in the RM model, particularly in the zero-shot setting. The evaluation on the HH-RLHF* dataset in the zero-shot scenario revealed unsatisfying performance, as indicated by an accuracy below 50%) As outlined in response to a previous question (Q2), RM (Vicuna-7B) employs a scalar reward generated solely through a fully connected layer. This approach, while chosen for its simplicity and efficiency, has inherent limitations, especially in the context of a shallow and weak signal. The model relies on the input question and answer, which may result in overfitting to training samples, particularly when evaluating with the complexities of the hh-rlhf* dataset in the zero-shot setting. In contrast, the proposed method outperforms vanilla RMs in both zero-shot and fine-tuning settings (as shown in Table 2). Moreover, it's essential to note that the task is not a traditional binary classification problem. RM (Vicuna-7B, Zero-shot) is designed to predict the preference score for both negative and positive pairs, then compute the correctness (acuracy) of the ranking order. We hope our response can address your questions.
>
> We hope the above can address your concerns.  We sincerely appreciate your time and thoughtful consideration. We look forward to any further discussion and appreciate your ongoing engagement with our work.

---

> > ### Comment · Reviewer_bKyt · 2023-11-21
> >
> > Thanks for your response. I sincerely appreciate your time in improving the overall clarity of your method.
> >
> > For Q3, If you output a random score for both negative and positive pairs, the probability that the score of a positive pair is higher than that of a negative pair is 50%. I'm still struggling to understand why the accuracy for RM using Vicuna-7B in a zero-shot setting is lower than 50%.
> >
> > I think another baseline could be considered. When using GPT-4 for labeling, we could also allow GPT to use the tool to provide intermediate results, engage in a chain of thought (COT), do the reasoning, and then give the final score. What advantages and disadvantages do you think your method has compared to GPT-4?
> >
> > Despite the questions raised by other reviewers, I believe the motivation of this work is commendable.

---

> > > ### Author Response · Authors · 2023-11-23
> > > **Urgent Request for Subsequent Discussion Before Closing**
> > >
> > > As the discussion period draws to a close very soon, we sincerely appreciate your time and expertise! We kindly request your continued engagement for further discussion and a re-review of our response. Your valuable insights contribute significantly to the refinement of our work.

---

> ### Author Response · Authors · 2023-11-22
> **Response to Reviewer bKyt Follow-Up (Part 1/2)**
>
> **Re: Q3 Follow-Up**. We appreciate the reviewer's continued engagement and would like to address the additional concern regarding the accuracy of RM using Vicuna-7B in a zero-shot setting.
>
> The observed lower performance is **primarily due to the challenges associated with domain generalization in vanilla RM**. Specifically, the **RM (Vicuna-7B) trained on tool-related data** using the vanilla RM objective (without the ability of tool use) may encounter **overfitting to the domain-specific data** (including *Calendar*, *Calculator*, *Weather*, *Code*, *Translator*, *Wikipedia*, and *Google Search*). Consequently, it achieves less than 50% accuracy (worse than random) on the HH dataset concerning human alignment, highlighting the vulnerability of vanilla RM to overfitting within the in-domain data. On the other hand, **Themis, which is also trained on tool-related data, attains a 55% accuracy** on the HH dataset (human alignment domain) in the zero-shot setting. This **underscores the exceptional out-of-domain generalization** in preference prediction, credited to the effective integration of tool use.
>
> Additionally, we would like to emphasize two key factors influencing zero-shot performance:
> 1. **Dataset Distribution and Domain Differences**: It is crucial to emphasize the substantial disparity in data distribution between our dataset, TARA, which is tool-related, and the HH-RLHF dataset, which focuses on human alignment with helpful and harmless content. The distinct nature of these datasets introduces challenges in achieving optimal zero-shot performance. The inherent differences in domain and data characteristics contribute to the observed lower accuracy in the zero-shot setting.
> 2. **Impact of Model Parameter Size**: The size of the model plays a significant role in zero-shot performance. Larger models, such as RM(Vicuna-7B), may exhibit a susceptibility to overfitting (easy to converge), as evidenced by the rapid convergence in just one training epoch (See Figure 3 Left). In contrast, a smaller model like RM (Bert-Large) requires more epochs to fit the reward data. The phenomenon of overfitting in a large model could explain the lower accuracy in the zero-shot setting, where the model may struggle to generalize effectively to unseen datasets. It's noteworthy that RM (Vicuna-7B) shows improved performance by +30% when finetuned on a smaller set of HH-RLHF instances (500), demonstrating the adaptability of the large model to relevant (in-domain) training data.
>
> To sum up, the challenges observed in zero-shot accuracy can be attributed to the significant differences in dataset distribution and the impact of model parameter size. We are committed to providing a more comprehensive understanding of these challenges and implementing necessary refinements to address them in the final version of our work.

---

> ### Author Response · Authors · 2023-11-22
> **Response to Reviewer bKyt Follow-Up (Part 2/2)**
>
> **Re: GPT-4 as the RM Baseline**. Thank you for your valuable suggestion. We appreciate the insightful idea of incorporating a ChatGPT or GPT-4 baseline in future experiments. Below, we outline the advantages and disadvantages of our proposed Themis model in contrast to GPT-4:
>
> |  | **RM (Themis)** | **RM (GPT-4)** |
> |---|---|---|
> | **Advantage** | (1) Ease of Application and Fine-tuning;  (2) Domain Specialization and Prompt-free Usage;  (3) Better Training/Inference Stability;  (4) Consistent Reward Format;  (5) Cost-effectiveness.  | (1) Larger model and strong capability;  (2) Strong generalization ability with prompt engineering.  |
> | **Disadvantage** | (1) Need data to fine-tune. | (1) High cost in time and money;  (2) Need prompt engineering;  (3) Unsupported RM fine-tuning;  (4) Closed-source. |
>
>
>
> **Advantages:**
> 1. **Ease of Application and Fine-tuning**:
>       * **Themis**: Can be easily applied and fine-tuned, making it compatible with traditional reward models.
>       * **GPT-4**: A closed-source model currently not supported by fine-tuning in the reward modeling context. It lacks the capability to add an additional fully-connected layer and conduct contrastive reward modeling training.
> 2. **Domain Specialization and Prompt-free Usage**:
>       * **Themis**: Facilitates easy domain specialization through fine-tuning, allowing adaptation to specific contexts. It also excels in prompt-free usage, automatically determining when and how to call pre-defined tools during inference.
>       * **GPT-4**: Requires complex prompt design and execution in each stage for domain-specific tasks. In particular, it relies on in-context learning ability and intricate prompt design for generating various stages, such as tool invocation and rationale stages. Importantly, prompt designs may need adjustment with GPT-4 updates.
> 3. **Better Training/Inference Stability**:
>       * **Themis**: Offers stability, avoiding recent downtimes associated with GPT-4/ChatGPT services due to increased calling burden. Themis is designed to be a reliable choice for model training.
>       * **GPT-4**: Suffers from various factors during training, such as OpenAI API services, network failures, and other potential disruptions.
> 4. **Consistent Reward Format**:
>       * **Themis**: Produces continuous scalar rewards through a fully connected layer, ensuring a consistent format.
>       * **GPT-4**: Generates rewards in natural language format, necessitating the formulation of rules to extract reward values from prompted results.
> 5. **Cost-effectiveness:**
>       * **Themis**: Serves as a reusable model that is significantly more cost-effective than GPT-4, providing an economical alternative for reward modeling tasks.
>       * **GPT-4**: Requires extra cost for API calling, adding to the overall expenses.
>
> **Disadvantages:**
> One notable disadvantage of Themis is the requirement for some data for fine-tuning. This contrasts with GPT-4, which can potentially leverage pre-existing knowledge without fine-tuning.
>
>
>
> We highly appreciate your valuable input, and we are committed to integrating these insights into our upcoming experiments and the refined version of our work. Thank you for acknowledging the motivation behind our tool-enhanced reward models.
>
> We hope these updates adequately address your concerns and kindly encourage you to reconsider our review score in light of these clarifications. Your contribution significantly enhances the quality of our research, and we thank you for your thoughtful feedback.

---

### Author Response · Authors · 2023-11-19
**Summary of Revisions**

Dear Reviewers and AC:

We express our sincere gratitude for your invaluable time and thoughtful feedback. Firstly, we are deeply appreciative of the recognition of the significance, innovation, and practicality of our method, as highlighted by Reviewer bKyt. Additionally, we acknowledge the positive remarks regarding the helpful features of reasoning and interpretability in our approach, as well as the commendation for the thoroughness of our experiments and implementation details, as noted by Reviewer wTHw. We also appreciate the comments from Reviewer zDVk, recognizing the novelty, timeliness, and interesting nature of our work, along with the acknowledgment of the logical and well-presented aspects of our submission.

To address your comments, we have implemented several key revisions in response to your constructive feedback. The changes are summarized below:

1. (For Reviewer bKyt, zDVk) Section 2.2 now explicitly states that our tool-augmented reward model (Themis) produces the scalar reward through a fully connected layer.
2. (For Reviewer wTHw) We have revised the data construction steps in Section 3.1 and Appendix B to provide greater clarity and transparency.
3. (For Reviewer zDVk) Discussions about the related work mentioned have been added to Section 5.1 to enhance the context and framing of our contribution.
4. (For Reviewer wTHw) The results of Themis (Vicuna-33B) have been added to Section 4, Table 1, as suggested.
5. (For Reviewer zDVk) A discussion about the challenges in dataset collection has been included in the Limitation section.
6. (For Reviewer bKyt, zDVk, wTHw) A table of hyperparameters $\alpha, \beta, \omega$ and training settings has been added to Appendix C.1.
7. (For Reviewer zDVk) Experimental results on standard reward datasets have been incorporated into Appendix C.3.
8. (For Reviewer wTHw) Strategies to avoid reward model interference by biases in tool invocations and two examples have been added to Appendix D.4.

We sincerely believe that these revisions address your concerns and contribute to the overall improvement of our work. We kindly encourage you to reconsider our review score in light of these updates. Your insights are invaluable, and we remain open to further discussion and refinement.
Thank you for your continued consideration. We genuinely look forward to contributing our work to the ICLR community.

Thank you very much,

Authors.

---

> ### Author Response · Authors · 2023-11-21
> **Urgent Request for Re-review and Discussion**
>
> Dear Reviewers and AC,
>
> We genuinely value the constructive comments and insightful suggestions you provided for our work. Recognizing the approaching end of the discussion period on **November 22nd**, we kindly request your valuable time and expertise for further discussion and re-review of our manuscript and response.
>
> In light of the feedback received and to reiterate the key contributions of our work:
> 1. **Introduction of the Tool-Enhanced Reward Model Framework**: We advance the domain of tool-augmented preference modeling by introducing the Themis framework. This innovative framework firstly harnesses the power of external tools to augment preference modeling in large language models (LLMs). It effectively addresses inherent limitations observed in conventional reward models (RMs) and provides transparency and interpretability by unveiling the inner workings of the RM's decision-making process.
> 2. **Creation of the Tool-Augmented Reward Modeling Dataset (TARA)**: We present the Tool-Augmented Reward Modeling Dataset (TARA), a novel dataset that includes comprehensive comparison data of human preferences and detailed tool invocation processes. Our commitment is to make this dataset publicly available, aiming to facilitate further research advancements in the field of tool-augmented preference modeling.
> 3. **Comprehensive Experimental Validation**: Our contributions are validated through extensive experimental evaluations conducted across eight diverse tasks, varying parameter scales, and experimental settings. Furthermore, we assessed the out-of-domain generalization of our approach using benchmarks like TruthfulQA and Retarded-bar (en) datasets. This evaluation was complemented by comprehensive RLHF investigations and in-depth analysis, highlighting the exceptional generalization capability and alignment ability of our method with humans and factual accuracy. The results of these experiments conclusively demonstrate the effectiveness of our approach in enhancing the performance of LLMs.
>
> We believe our work can accelerate the use of tools in human alignment, providing more accurate and informative feedback to guide LLM fine-tuning. Additionally, we reaffirm our commitment to the open-source community by planning to release our code, data, and models, fostering collaboration and further research in this area.
>
> We kindly urge you to participate in the ongoing discussion and provide any additional insights or clarifications you may have. Your expertise is invaluable to us, and we believe your input will significantly contribute to the improvement of our work.
>
> Thank you very much for your time and consideration. We look forward to hearing from you soon.
>
>
> Authors

---

### Meta-Review · Area_Chair_ic2Y · 2023-12-11

**Metareview:**

### Summary
Traditional reward models (RMs) have difficulty handling complex tasks requiring external tools, such as arithmetic computation or code execution. This paper proposes a tool-augmented preference-based reward modeling approach, **Themis**, to address these limitations. Themis integrates external tools like a calculator into RMs, allowing them to interact with external sources and make task-specific tool engagement and reasoning traces. The authors evaluated Themis on various tasks and found that it significantly outperforms traditional RMs. They also provide a comprehensive tool-related RM dataset (TARA) for further research.

### Decision

The paper proposes an exciting and reasonable way to improve the reward models. The results and the improvements over the compared baselines are impressive. The paper reports a 7.3% improvement in TruthfulQA over Gopher 280B. The paper is easy to read. All the reviewers unanimously agreed to accept this paper, considering the strengths of the proposed approach. The authors have done an excellent job of addressing the concerns raised by the reviewers during the rebuttal period. Two of the reviewers assigned a score of 8 out of 10. This paper deserves to be represented at ICLR as a spotlight paper.

**Justification For Why Not Higher Score:**

The idea is exciting and novel, and the experiments are well-executed. However, the writing quality could be improved. A more detailed study of the robustness of the tool-augmented reward models would be valuable or providing insights on improving our theoretical understanding would make this paper definitely oral-worthy.

**Justification For Why Not Lower Score:**

The paper is well-written. The authors did a tremendous job replying to the reviews during rebuttal. All the reviewers agree that the paper should be accepted. The idea is simple but effective and it is not surprising that it works well. The paper deserves recognition as a spotlight.

---

### Decision · Program_Chairs · 2024-01-16

Accept (spotlight)